# DYNAMIC MULTI-CHANNEL EEG GRAPH MODELING FOR TIME-EVOLVING BRAIN NETWORK

## ABSTRACT

We describe a novel dynamic graph neural network (GNN) approach for seizure detection and prediction from multi-channel Electroencephalography (EEG) data that addresses several limitations of existing methods. While deep learning models have achieved notable success in automating seizure detection, static graph-based methods fail to capture the evolving nature of brain networks, especially during seizure events. To overcome this, we propose `EvoBrain`, which uses a *time-then-graph* strategy that first models the temporal dynamics of EEG signals and graphs, and then employs GNNs to learn evolving spatial EEG representations. Our contributions include (a) a theoretical analysis proving the expressivity advantage of *time-then-graph* over other approaches, (b) a simple and efficient model that significantly improves AUROC and F1 scores compared with state-of-the-art methods, and (c) the introduction of dynamic graph structures that better reflect transient changes in brain connectivity. We evaluate our method on the challenging early seizure prediction task. The results show improved performance, making `EvoBrain` a valuable tool for clinical applications. The source code is available at: https://anonymous.4open.science/r/EvoBrain-FBC5

## 1 INTRODUCTION

Seizure affects 60 million of the population worldwide, and approximately 40% of patients are drug-resistant that available medications cannot effectively control (WHO, 2024). Clinical analysis, including detection and prediction, is vital as regards intervention and surgical treatment (Surges et al., 2021). Electroencephalography (EEG) reveals the activities of billions of neurons in multi-channel recordings and is the preferred tool for analyzing seizures. Despite promising clinical outcomes, these advances impose massive burdens on clinicians, who need to review recordings made over many days to identify seizure events. Due to the lack of validated EEG biomarkers, current gold-standard detection still requires video monitoring, i.e., v-EEG (Zhang et al., 2022), which is costly and only available in tertiary hospitals, hindering timely diagnosis and broader seizure research.

Deep learning models achieved noteworthy results as regards automating seizure detection using EEGs (Khan et al., 2018; Burrello et al., 2020; Eldele et al., 2021; Chen et al., 2022; Yang et al., 2023b; Yi et al., 2023; Jiang et al., 2024). In particular, graph neural networks (GNNs) (Tang et al., 2022; Cai et al., 2023), which leverage non-Euclidean spatial information in the brain, have shown potential for identifying abnormal connections across several channels, serving as more effective seizure markers (Li et al., 2021a). Researchers typically construct graph-based EEG representations, where nodes indicate to channels and edges represent connections between them, often defined by node-node similarity. Through graph learning, these methods capture interactions and detect abnormal patterns among brain regions. However, most existing methods are static graphs, embedding raw EEGs as node features (Chen et al., 2022). This hinders the ability to fully capture the temporal dynamics of brain activity, which are crucial for seizure analysis (D. V. Fallani et al., 2014).

To tackle this, temporal models integrated with GNNs and known as dynamic GNNs have emerged as a promising way of learning evolving patterns in EEGs explicitly (Jia et al., 2020; Feng et al., 2022). Methods for seizure research can be categorized into *graph-then-time* and *time-and-graph* approaches, as shown in Figure 1. The *graph-then-time* approach represents EEGs as a sequence of graph snapshots (Cai et al., 2023). GNNs are applied to each snapshot independently, learning channel correlations at each time step. The outputs are concatenated and embedded in an RNN-based

sequential model to learn temporal dynamics. The *time-and-graph* approach proposes an additional recurrent GNN to learn interactions between EEG snapshots. It updates and evolves node features based on the RNN cell output activated by the previous graph snapshots. While state-of-the-art (SOTA) performance has been demonstrated (Tang et al., 2022), only a few studies have included further investigations of dynamic GNNs in seizure modeling, and some challenges may remain.

1. **Inadequate learning of temporal dynamics.** The independent GNNs in *graph-then-time* still provide static graphs. They represent information at single time steps without accounting for interactions between different time steps. While recurrent GNNs in *time-and-graph* capture graph interactions, they rely on the independent initialization of the EEG graphs. This biases the model toward graph information from earlier steps, limiting its ability to capture dynamic changes.

2. **Empirical modeling.** There has been little theoretical analysis of dynamic GNNs when modeling seizures. While existing studies (Tang et al., 2022; Ho & Armanfard, 2023) offer valuable insights, the optimal modeling strategy for combining temporal and graph-based representations, as well as graph aggregation techniques for multi-channel EEG, remain poorly understood.

3. **Static and fixed graph structure.** While these methods are labeled as "dynamic," they provide static graph structures. The construction is predefined using channel correlations in the first snapshot and remains fixed across time. This setting means only the temporal aspect of the nodes is considered dynamic. Such fixed structures fail to represent the constantly changing nature of brain networks (Bassett & Sporns, 2017; Li et al., 2021a).

this paper investigates a new dynamic GNNs approach, *time-then-graph* (Gao & Ribeiro, 2022), for modeling multi-channel EEGs. This leads us to `EvoBrain`, which effectively learns `Evo`lving, dynamic characteristics in `Brain` networks for accurate seizure detection and prediction. `EvoBrain` first sequentially represents the temporal evolution of nodes and edges independently. Then, a GCN is used to learn graph representations using sequential representations and their temporal interactions. Technically, ( 1 ) we explore the expressivity of three dynamic GNNs approaches in modeling EEG dynamics and theoretically prove that *time-then-graph* has a potential expressivity advantage over the other methods. ( 2 ) We propose dynamic graph structures that represent brain connectivity in a series of snapshot-dependent graphs. When incorporated with temporal models, the dynamic graphs provide insights into how brain networks vary. **Contributions**:

- **Theoretical EEG modeling analysis**. We are the first to theoretically analyze different dynamic GNNs approaches from a node representation perspective. The work of Gao & Ribeiro (2022) provides a general proof for unattributed and edge-only graphs. However, we analyze the nessarity of dynamic GNNs at node-level, since the node features and node similarity measures are key factors in determining EEG graph construction (Ho & Armanfard, 2023). We provide a foundation for designing dynamic GNNs that effectively represents brain networks.

- **Simple yet new SOTA performance**. `EvoBrain` is a simple GRU-GCN architecture consisting of only two GRUs and one GCN model. Despite its simplicity it achieves up to 8.5% and 15% improvements in AUROC and F1 scores, respectively, compared with the SOTA seizure detection baseline. Due to its simple architecture design, `EvoBrain` is 23× faster than the SOTA *time-and-graph* method. More advanced model architectures can be easily integrated into `EvoBrain`.

- **Dynamic graph structure**. We propose dynamic graph structures that incorporate temporal EEG graph modeling. The structures accurately reflect the true nature of brain networks, where connectivity between regions fluctuates rapidly. The experiments confirm that simply using dynamic structures can improve performance, even for *time-and-graph* and *graph-then-time* approaches.

- **Early prediction task**. Unlike most detection approaches (Eldele et al., 2021; Cai et al., 2023; Ho & Armanfard, 2023), we evaluate the more challenging task of seizure prediction, which aims to identify the preictal state before seizures. This is critical for early intervention in clinical settings, and `EvoBrain` consistently maintains performance, with a 13.8% improvement in AUROC.

## 2 RELATED WORK

**Automated Seizure analysis.** The automated detecting or prediction of seizures has been a long-standing challenge (Zhang et al., 2024b; Zheng et al., 2024). Deep learning has shown great achievements in automating EEG feature extraction and detection, using convolutional neural networks (CNNs) (Fukumori et al., 2022a; Ahmedt-Aristizabal et al., 2020; Asif et al., 2020; Saab et al.,

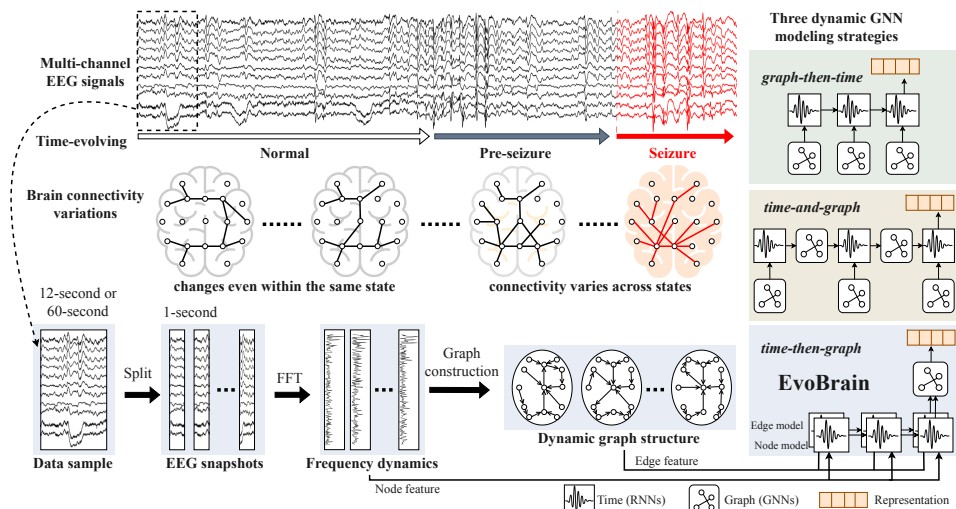

Figure 1: The brain network evolves over time, and changes occurring during seizures and immediately before them in the pre-seizure phase, especially within specific zones, are clinically important. These changes are captured using dynamic graphs derived from multi-channel EEG signals. Three dynamic GNNs approaches to modeling these signals are shown: *graph-then-time*, *time-and-graph*, and *time-then-graph*. Here `EvoBrain` is built based on *time-then-graph* approach.

2020), RNN-based models (Fukumori et al., 2022b; Ahmedt-Aristizabal et al., 2020; Rasheed et al., 2021), Transformers (Eldele et al., 2021; Yang et al., 2023b; Jiang et al., 2024; Yi et al., 2023), and brain-inspired models (Rich et al., 2020; Burelo et al., 2022; Chen et al., 2023; Costa et al., 2024).

**Spatial Relationships in EEG networks.** A seizure is fundamentally a network disease, and detection typically relies on the ability to determine abnormalities in EEG channels (Burns et al., 2014a; Li et al., 2021a; Rolls et al., 2021). Many multi-channel methods have been proposed for capturing spatial information in channels (Jiang et al., 2024; Yi et al., 2023; Zhang et al., 2023; Mohammadi Foumani et al., 2024). Among them, recent studies have proposed GNNs to capture further the non-Euclidean structure of EEG electrodes and the connectivity in brain networks (Covert et al., 2019; Sun et al., 2021; Li et al., 2022; Chen et al., 2022; Klepl et al., 2022; Demir et al., 2022). These methods form EEGs as graphs, embedding each channel into the nodes and learning spatial graph representations (Demir et al., 2021; Ho & Armanfard, 2023). However, they do not explicitly model temporal relationships, relying instead on convolutional filters or conventional linear projections for node embeddings.

**Dynamic GNNs for EEG Modeling.** Dynamic GNN is effective in learning temporal graph dynamics, achieving promising results in tasks such as dynamic link prediction (Tian et al., 2024), node classification (Zhang et al., 2024a), and graph clustering (Liu et al., 2024). Recently, two studies have focused on dynamic GNNs desired to enhance temporal dynamic and graph representations for EEG-based seizure modeling. Tang et al. (2022) proposes a *time-and-graph* model , which uses frequency features from FFT as node features and applies GNN and RNN processing simultaneously for each sliding window. Cai et al. (2023) adopts a *graph-then-time* model that combines GCN and RNN for seizure detection. However, these studies construct static graphs with fixed structures across temporal learning. Hou et al. (2022) propose *time-then-graph* model, BiLSTM-GCNNet, which uses RNNs to construct static graph from node feature. GRAPHS4MER (Tang et al., 2023) is also proposed as a *time-then-graph* method. This work effectively learns dynamic graphs using an intermediate graph structure learning model. However, its input for graph structure learning is still based on the Euclid distance or similarity of the entire data sample (e.g., 12 or 60 seconds) rather than individual snapshots. Our work differs by defining dynamic graph structures that more effectively capture the temporal evolution of brain connectivity in EEGs.

## 3 DYNAMIC GNN MODELING ANALYSIS IN EEG

### 3.1 PROBLEM FORMULATION

**Notations.** We define an EEG **X** with $N$ channel and $T$ snapshots as a graph $\mathcal{G} = (\mathbf{A}, \mathbf{X})$, where $\mathbf{A} = (\mathcal{V}, \mathcal{E})$ and $\mathbf{A} \in \mathbb{R}^{N \times N \times T}$ is the adjacency matrix. $\mathcal{V}$ and $\mathcal{E}$ represent the channels (i.e., nodes) and edges, respectively. Notably, existing work construct **A** as fixed across $T$ meaning that all EEG snapshots share the same graph structure and only the node features $H$ are computed iteratively at each snapshot. In this paper, each edge $e_{i,j,t} \in \mathcal{E}$ represents pairwise connectivity between channels $v_i$ and $v_j$, where $i, j \in N$. The feature vector $x_{i,t} \in \mathbb{R}^d$ captures the electrical activity of $i$-th channel during the EEG snapshot at time step $t$. If $e_{i,j,t}$ exists, $a_{i,j,t}$ is a non-zero value. $a_{i,j,t} \in \mathbb{R}$ quantifies the connectivity strength between two channels for each snapshot. To represent temporal EEG graphs, we define the embedding of node $v_i$ at time step $t$ as $h_{i,t}^{node} \in \mathbb{R}^k$, , which captures both the spatial connectivity information from the adjacency matrix **A** and the temporal dynamics from previous embeddings. The embedding of edge $e_{i,j,t}$, denoted as $h_{i,j,t}^{edge} \in \mathbb{R}^l$, captures the temporal evolution of channel connectivity, reflecting changes in brain networks.

**Problem (Dynamic GNN Expressivity in EEG Modeling.)** We aim to investigate the expressivity of temporal graph representation methods, including *graph-then-time*, *time-and-graph*, and *time-then-graph*, in the context of dynamic EEG graph analysis. Brain networks in different states could manifest as distinct graph structures, as shown in Figure 1. Clinically, abnormal EEG channel connectivity may serve as seizure markers (Li et al., 2021a). We define **Expressivity Analysis** as a *graph isomorphism* problem (Xu et al., 2019), where non-isomorphic EEG graphs represent different brain states, enabling the model to effectively distinguish between seizure and non-seizure graphs.

### 3.2 EXPRESSIVITY ANALYSIS FOR DYNAMIC EEG GRAPHS

In the following analysis, we examine expressivity in node representation degree, as the EEG graph structure is often based on node correlations (correlation graph (Tang et al., 2022)). We employ the 1-Weisfeiler-Lehman (1-WL) GNNs for analyzing graph isomorphism (provided in Appendix A). The GNNs that can effectively differentiate non-isomorphic graphs have high expressiveness.

**Lemma 1.** *[Necessity of Node Representations]* **Edges alone** (Gao & Ribeiro, 2022) **are insufficient** *to uniquely distinguish certain temporal EEG graphs. Specifically, there exist pairs of temporal EEG graphs that have identical edge features across all time steps but different node features, making them indistinguishable based solely on edge representations. Therefore,* **incorporating node representations is necessary** *to achieve full expressiveness in EEG graph classification tasks.*

*Proof.* Given $\mathcal{G}^{(1)}$ and $\mathcal{G}^{(2)}$, with the same sets of $\mathcal{V}$ and $\mathcal{E}$, for $\forall\, t$, the edge features satisfy: $\mathcal{A}_{i,j,t}^{(1)} = \mathcal{A}_{i,j,t}^{(2)} \quad \forall (v_i, v_j) \in \mathcal{E}, \forall t \in \{1, 2, \ldots, T\}$. However, suppose there exists at least one node $v_k \in \mathcal{V}$ and one time step $t'$ such that: $\mathcal{X}_{k,t'}^{(1)} \neq \mathcal{X}_{k,t'}^{(2)}$. Since $\mathcal{A}_{i,j,t}^{(1)} = \mathcal{A}_{i,j,t}^{(2)}$, any GNN architecture that relies solely on edge features will produce identical embeddings for $\mathcal{G}^{(1)}$ and $\mathcal{G}^{(2)}$, $\forall t$. $\square$

**Lemma 2.** *[Expressiveness with Node and Edge Representations] When both node and edge representations are incorporated, a GNN can uniquely distinguish any pair of temporal EEG graphs that differ in either node features or edge features at any time step, provided the GNN is sufficiently expressive (e.g., 1-WL GNN). Details of Lemma 2 can be found in Appendix B.1.*

**Definition 1** (Graph-then-time). *This approach first apply GNNs to learn spatial, graph information at each $t$ independently, followed by the temporal processing (e.g., by RNNs) of the resulting node embeddings. This approach prioritizes spatial relationships before incorporating the temporal dynamics across EEG snapshots. The formal definition is given as:*

$$\mathbf{H}_{i,t} = Cell\left(\left[GNN_{in}^L(\mathbf{X}_{:,t}, \mathbf{A}_{:,:,t})\right]_i, \mathbf{H}_{i,t-1}\right) \tag{1}$$

*Here, $\mathbf{H}_{i,t}$ denotes the embedding of node $i \in \mathcal{V}$ at time $t$. $GNN_{in}^L(\mathbf{X}_{:,t}, \mathbf{A}_{:,:,t})$ denotes a graph learning on the current snapshot. The learned embeddings at time step $t-1$, $\mathbf{H}_{i,t-1}$, are then passed into the RNN cell, or other recurrent unit, to capture the temporal dependencies.*

**Definition 2** (Time-and-graph). *This approach alternately processes time and graph components, applying GNNs to each EEG snapshot, as formally defined by:*

$$\mathbf{H}_{i,t} = Cell\bigg( \Big[ GNN_{in}^L(\mathbf{X}_{:,t}, \mathbf{A}_{:,:,t}) \Big]_i, \quad \Big[ GNN_{rc}^L(\mathbf{H}_{:,t-1}, \mathbf{A}_{:,:,t}) \Big]_i \bigg), \quad \mathbf{Z} = \mathbf{H}_{i,T}, \quad \forall i \in \mathcal{V} \quad (2)$$

*where $\mathbf{H}_{i,t}$ is the final representations of temporal node $i \in V$ at time $1 \le t \le T$. We initialize $H_{i,0} = 0$ for $\forall\, i$. $GNN_{in}^L$ encodes each $\mathbf{X}_{:,t}$ while $GNN_{rc}^L$ encodes representations from historical snapshots $\mathbf{H}_{:,t-1}$, and Cell embeds evolution of those graph representations. For an arbitrary temporal EEG graph, the last step output $\mathbf{H}_{i,T}$ is considered the final representation of node $i \in \mathcal{V}$.*

**Definition 3** (Time-then-graph). *This approach first models the evolution of node and edge attributes over time and then applies a GNN to the resulting static graph for final representation:*

$$\mathbf{H}_i^{node} = RNN^{node}(\mathbf{X}_{i,\le T}), \forall i \in \mathcal{V}, \quad \mathbf{H}_{i,j}^{edge} = RNN^{edge}(\mathbf{A}_{i,j,\le T}), \forall (i,j) \in \mathcal{E}, \quad (3)$$

$$\mathbf{Z} = GNN^L(\mathbf{H}^{node}, \mathbf{H}^{edge})$$

*time-then-graph represents the evolution of $\mathbf{H}^{node}$ and $\mathbf{H}^{edge}$ using two sequential models $RNN^{node}$ and $RNN^{edge}$, resulting in a new (static) graph, which is then encoded by a $GNN^L$.*

**Lemma 3.** *[graph-then-time $\npreceq$ time-and-graph ] time-and-graph is strictly more expressive than graph-then-time representation family on $\mathbb{T}_{n,T,\theta}$ as long as we use 1-WL GNNs.*

*Proof.* By Definition 1, $\mathbf{H}_{i,t-1}$ is passed without this additional GNN (i.e., $GNN_{rc}^L(\cdot)$) to learn interactions between EEG snapshots. This results in a simpler form of temporal representation compared to *time-and-graph*: $\mathbf{H}_{i,t-1} \subseteq \big[ GNN_{rc}^L(\mathbf{H}_{:,t-1}, \mathbf{A}_{:,:,t}) \big]_i$. *graph-then-time* is a strict subset of *time-and-graph* in terms of expressiveness. $\square$

**Lemma 4.** *[time-and-graph $\npreceq$ time-then-graph ] time-then-graph is strictly more expressive than time-and-graph representation family on $\mathbb{T}_{n,T,\theta}$, as time-then-graph outputs different representations, while time-and-graph does not.*

*time-then-graph* learn node and edge features across time steps to capture temporal dependencies. This is done by encoding the temporal adjacency matrices $\mathbf{A}_{:,:,\le t}$ and node features $\mathbf{X}_{:,\le t}$ together, enabling the model to distinguish between graphs with distinct temporal structures. However, *time-and-graph* handles each time step independently, leading to identical representations across time.

**Theorem 1.** *[Temporal EEG Graph Expressivity] Based on Lemmas 3 and 4, we conclude that graph-then-time is strictly less expressive than time-and-graph, and time-and-graph is strictly less expressive than time-then-graph on $\mathbb{T}_{n,T,\theta}$, when the graph representation is a 1-WL GNN:*

$$\text{graph-then-time} \npreceq_{\mathbb{T}_{n,T,\theta}} \text{time-and-graph} \npreceq_{\mathbb{T}_{n,T,\theta}} \text{time-then-graph.} \quad (4)$$

In Appendix B.2, we prove Lemma 4 using both node and edge representation perspectives, based on Lemma 2 to hold Theorem 1, Notably, we provide a synthetic EEG task where any *time-and-graph* representation fails, while a *time-then-graph* approach succeeds.

## 4 EVOBRAIN

Based on the above analysis, this section presents our `EvoBrain`, which is built on top of *time-then-graph*, and we propose a dynamic graph construction method to represent temporal EEG structures.

### 4.1 DYNAMIC BRAIN GRAPH STRUCTURE

**Dynamic EEG Graph** Instead of constructing a single static graph from the entire EEG recording, we propose to construct EEG graph for each snapshot. We first segment an EEG epoch into short time durations (i.e., snapshots) at regular intervals and compute channel correlations to construct a sequence of graph structures. Specifically, for the $t$-th snapshot, we define the edge weight $a_{i,j,t}$ as the weighted adjacency matrix $\mathbf{A}$, computed as the absolute value of the normalized cross-correlation between nodes $v_i$ and $v_j$. To prevent information redundancy and create sparse graphs, we rank the correlations among neighboring nodes and retain only the edges with the top-$\tau$ highest correlations.

$$a_{i,j,t} = |x_{i,t} * x_{j,t}|, \text{ if } v_j \in \mathcal{N}(v_i), \text{ else } 0,$$

where $x_{i,:,t}$ and $x_{j,:,t}$ represent $v_i$ and $v_j$ channels of $t$-th EEG snapshot. $*$ denotes the normalized cross-correlation operation. $\mathcal{N}(v_i)$ denotes the set of top-$\tau$ neighbors of $v_i$ with relative higher correlations. After computing this for $T$ snapshots, we obtain a sequence of directed, weighted EEG graph $G$ to represent brain networks at different time points. In other words, the dynamic nature of the EEG is captured by the evolving structure of these graphs over time.

## 4.2 Dynamic GNN in Time-Then-Graph Framework

Following the *time-then-graph* approach outlined in Definition 3, we propose a GRU-GCN model, where GRUs learn the temporal evolution of node and edge attributes independently, followed by a GCN to capture spatial dependencies across electrodes in a static graph. This method effectively captures the temporal and spatial dynamics inherent in EEG data for seizure detection and prediction. Notably, the model input is not the raw EEG signals but their **frequency spectrum** representation. Here, clinical seizure analysis aims to identify specific frequency oscillations and waveforms, such as spikes(Khan et al., 2018). To effectively capture such features, we apply a fast Fourier transform (FFT) to each EEG snapshot, retaining the log amplitudes of the non-negative frequency components, following prior studies (Covert et al., 2019; Asif et al., 2020; Tang et al., 2022). The EEG snapshots are then normalized using z-normalization across the training set. Consequently, an EEG frequency representation with a sequence of snapshots is formulated as $\mathbf{X} \in \mathbb{R}^{N \times d \times T}$, serving $N$ node initialization and dynamic graph construction.

**Temporal Modeling with GRUs** Given a dynamic EEG graphs $G$, for each channel (node) $i \in \mathcal{V}$, the node attribute sequence $\{\boldsymbol{X}_{i,t}\}_{t=1}^{T}$ is processed by a GRU to obtain a hidden node representation. This captures the temporal-frequency dependencies across EEG snapshots. Similarly, for each channel connectivity (edge) $(i,j) \in \mathcal{E}$, the edge attribute sequence $\{\boldsymbol{A}_{ij,t}\}_{t=1}^{T}$ is processed by another GRU to obtain a edge hidden representation. Since each edge is defined based on a short-time EEG snapshot, this edge representation learns how EEG channel connectivity evolves across time/snapshots. These processes can be formulated as:

$$\mathbf{h}_i^{\text{node}} = \text{GRU}^{\text{node}}\big(\{\boldsymbol{X}_{i,t}\}_{t=1}^{T}\big), \forall i \in \mathcal{V}, \quad \mathbf{h}_{ij}^{\text{edge}} = \text{GRU}^{\text{edge}}\big(\{\boldsymbol{A}_{ij,t}\}_{t=1}^{T}\big), \forall (i,j) \in \mathcal{E}. \quad (5)$$

We express the GRU updates for both nodes and edges in a unified manner. For each element $e$ (which can be a node $i$ or an edge $(i,j)$), the GRU updates at each time step $t$ are defined as:

$$\begin{aligned}
\mathbf{r}_t^e &= \sigma\big(\mathbf{W}_r \mathbf{x}_t^e + \mathbf{U}_r \mathbf{h}_{t-1}^e + \mathbf{b}_r\big), \quad \mathbf{z}_t^e = \sigma\big(\mathbf{W}_z \mathbf{x}_t^e + \mathbf{U}_z \mathbf{h}_{t-1}^e + \mathbf{b}_z\big), \\
\mathbf{n}_t^e &= \tanh\big(\mathbf{W}_n \mathbf{x}_t^e + \mathbf{U}_n\big(\mathbf{r}_t^e \odot \mathbf{h}_{t-1}^e\big) + \mathbf{b}_n\big), \quad \mathbf{h}_t^e = (1 - \mathbf{z}_t^e) \odot \mathbf{n}_t^e + \mathbf{z}_t^e \odot \mathbf{h}_{t-1}^e,
\end{aligned} \quad (6)$$

where $\mathbf{x}_t^e$ is the input feature vector of element $e$ at time $t$, $\mathbf{h}_t^e$ is the hidden state at time $t$, $\sigma$ denotes the sigmoid function, $\odot$ represents element-wise multiplication, and $\mathbf{W}_*, \mathbf{U}_*, \mathbf{b}_*$ are learnable parameters (separate for nodes and edges). Finally, we obtain the final hidden states $\mathbf{h}_i^{\text{node}} = \mathbf{h}_T^i$ and $\mathbf{h}_{ij}^{\text{edge}} = \mathbf{h}_T^{(i,j)}$, which encapsulate the temporal evolution of node and edge attributes.

**Spatial Modeling with GCNs** We hence construct a new graph $\mathcal{G} = (\mathcal{V}, \mathcal{E})$ where the nodes and edges are embedded with their respective temporal representations, $\mathbf{h}_i^{\text{node}}$ and $\mathbf{h}_{ij}^{\text{edge}}$. We then adapt a GCN to learn spatial dependencies. These graph embeddings capture the temporal evolution of EEG snapshots, with each snapshot reflecting the brain state at that particular time. This GCN learning can thus be viewed as summarizing the overall graph interactions and dynamics, defined as follows:

$$\mathbf{Z} = \mathrm{GNN}^L\big(\{\mathbf{h}_i^{\mathrm{node}}\}_{i\in\mathcal{V}}, \{\mathbf{h}_{ij}^{\mathrm{edge}}\}_{(i,j)\in\mathcal{E}}\big). \tag{7}$$

Each layer of the GCN updates the node embeddings by aggregating information from neighboring nodes and edges. Specifically, the node embeddings are updated as follows:

$$\mathbf{h}_i^{(l+1)} = \sigma\left(\sum_{j\in\mathcal{N}(i)} f_{\mathrm{edge}}\big(\mathbf{h}_{ij}^{\mathrm{edge}}\big) \odot \mathbf{h}_j^{(l)}\mathbf{\Theta}^{(l)}\right), \tag{8}$$

where $\mathbf{h}_i^{(l)}$ is the embedding of node $i$ at layer $l$, $\mathcal{N}(i)$ denotes the neighbors of node $i$, $f_{\mathrm{edge}}$ is a function mapping edge embeddings to scalar weights or messages, $\mathbf{\Theta}^{(l)}$ is the learnable weight matrix at layer $l$, and $\sigma$ is an activation function (e.g., ReLU).

Afterward, we apply max pooling over the node embeddings, i.e., $\mathbf{h}_i^{(L)}$, followed by a fully connected layer and softmax activation for seizure detection and prediction tasks.

## 5 EXPERIMENTS

### 5.1 EXPERIMENTAL SETUP

**Tasks.**  In this study, we focus on two tasks: seizure detection and seizure early prediction.

- **Seizure detection** is framed as a binary classification problem, where the goal is to distinguish between seizure and non-seizure EEG segments, termed epochs. This task serves as the foundation for automated seizure monitoring systems.

- **Seizure early prediction** is the more challenging and clinically urgent task. It aims to predict the onset of an epileptic seizure before it occurs. Researchers typically frame this task as a classification problem (Burrello et al., 2020; Batista et al., 2024), where the goal is to distinguish between pre-ictal EEG epochs and the normal state. Accurate classification enables timely patient warnings or preemptive interventions, such as electrical stimulation, to prevent or mitigate seizures.

**Datasets.**  We used the Temple University Hospital EEG Seizure dataset v1.5.2 (**TUSZ**) (Shah et al., 2018) to evaluate `EvoBrain`. Description can be found in Appendix G. TUSZ is the largest public EEG seizure database, containing 5,612 EEG recordings with 3,050 annotated seizures. Each recording consists of 19 EEG channels. A key strength of TUSZ is its diversity, with data collected over different time periods, using various equipment, and covering a wide age range of subjects.
Additionally, we used the smaller CHB-MIT dataset, which consists of 844 hours of 22-channel scalp EEG data from 22 patients, including 163 recorded seizure episodes.
*Preprocessing.* For early prediction task, we defined the *one-minute period* before a seizure as the preictal phase, implying the ability to predict seizures up to one minute in advance. Detailed description can be found in Appendix F.

**Baselines.**  We selected two dynamic GNNs studies as baselines: EvolveGCN-O (Pareja et al., 2020), which follows the *graph-then-time* approach, a *time-and-graph* work, DCRNN (Tang et al., 2022), and *time-then-graph* approach, GRAPHS4MER (Tang et al., 2023). We included a benchmark Transformer baseline, BIOT (Yang et al., 2023a), which captures temporal-spatial information for various EEG tasks. We also evaluated LSTM (Hochreiter & Schmidhuber, 1997) and CNN-LSTM (Ahmedt-Aristizabal et al., 2020), as referenced in (Tang et al., 2022), Support Vector Machine (SVM) and Random Forest to assess the effectiveness of our temporal-graph learning.

**Metrics.**  We used Area Under the Receiver Operating Characteristic curve (AUROC) and F1 score as evaluation metrics. AUROC considers various threshold scenarios, providing an overall measure of the model's ability to distinguish between classes across a range of decision boundaries. F1 score focuses on selecting the best threshold by balancing precision and recall, highlighting the model's performance at its most optimal point for a specific classification task.

**Model training.** Training for all models was accomplished using the Adam optimizer (Kingma & Ba, 2014) in PyTorch on NVIDIA A6000 GPU and Xeon Gold 6258R CPU. During training, we performed data augmentation. Details are provided in Appendix D.

## 5.2 RESULTS

Table 1: Performance comparison of TUSZ dataset on seizure detection and prediction for 12s and 60s. The **best** and second best results are highlighted.

| Method | Type | Detection | | | | Prediction | | | |
|---|---|---|---|---|---|---|---|---|---|
| | | 12s | | 60s | | 12s | | 60s | |
| | | AUROC | F1 | AUROC | F1 | AUROC | F1 | AUROC | F1 |
| SVM | - | 0.765 | 0.369 | 0.721 | 0.390 | 0.562 | 0.312 | 0.561 | 0.312 |
| Random Forests | - | 0.778 | 0.354 | 0.737 | 0.384 | 0.563 | 0.352 | 0.547 | 0.327 |
| LSTM | - | 0.794 | 0.381 | 0.721 | 0.390 | 0.568 | 0.353 | 0.553 | 0.387 |
| CNN-LSTM | - | 0.754 | 0.354 | 0.680 | 0.329 | 0.621 | 0.389 | 0.528 | 0.314 |
| BIOT (Yang et al., 2023a) | - | 0.726 | 0.320 | 0.637 | 0.256 | 0.540 | 0.390 | 0.576 | 0.390 |
| EvolveGCN (Pareja et al., 2020) | *graph-then-time* | 0.757 | 0.343 | 0.655 | 0.334 | 0.622 | 0.437 | 0.511 | 0.356 |
| DCRNN (Tang et al., 2022) | *time-and-graph* | 0.817 | 0.415 | 0.802 | 0.431 | 0.626 | 0.389 | 0.621 | 0.448 |
| GRAPHS4MER (Tang et al., 2023) | *time-then-graph* | 0.833 | 0.413 | 0.765 | 0.439 | 0.648 | 0.440 | **0.651** | 0.370 |
| **EvoBrain (Ours)** | *time-then-graph* | **0.869** | **0.506** | **0.832** | **0.443** | **0.679** | **0.473** | 0.631 | **0.452** |

**Main results.** Table 1 presents a performance comparison of seizure detection and prediction for the TUSZ dataset using various models over 12-second and 60-second windows. `EvoBrain` consistently outperforms baselines. For seizure detection in 12-second window, `EvoBrain` improves over two dynamic GNNs baselines. `EvoBrain` shows a 15% increase in AUROC compared with EvolveGCN (0.756 → 0.870) and a 7% increase compared to DCRNN (0.813 → 0.870). It also improves the F1 score by 43.6% (0.351 → 0.504). These results highlight that even with a simple architecture design, `EvoBrain` shows significant improvements, particularly in F1 scores across both tasks and window lengths, supporting our analysis and conclusion of Theorem 1 in Section 3.

Figure 2 shows the ROC curves results comparing `EvoBrain` with other dynamic GNN approaches. In subfigure (a), for the TUSZ dataset, `EvoBrain` achieves an AUC of 0.87, outperforming DCRNN (0.81) and EvolveGCN (0.76). Our ROC curve is positioned higher, indicating a stronger ability to differentiate between seizure and non-seizure events. In subfigures (b) for the CHB-MIT dataset, `EvoBrain` achieves an AUC of 0.90, significantly higher than the 0.81 and 0.59 of *time-and-graph* and *graph-then-time* approaches, respectively. The results show the effectiveness and discriminative ability of *time-then-graph* for identifying seizures.

**Dynamic graph structure evaluation.**

Figure 3 shows the effectiveness of our proposed dynamic graph structure compared to the static graph structures commonly used in existing works. The blue bar shows the performance of the original static graph structure used in EvolveGCN and DCRNN, while the orange bar represents the results when the static graph is replaced with our dynamic graph structures. As seen, the improvements are **not limited** to our *time-then-graph* method but also enhance the performance of all dynamic GNNs approaches. The figure highlights the effectiveness and necessity of dynamic graphs in capturing brain dynamics. The results imply that modeling temporal dynamics in EEGs should incorporate various channel connectivity or structural information.

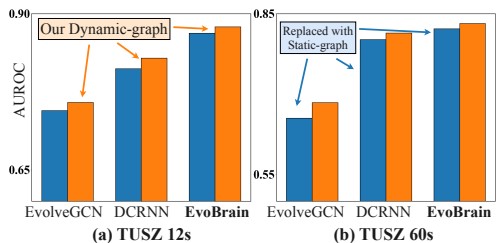

Figure 2: ROC curve results for the 12-second seizure detection task on two datasets.

Figure 3: Comparison of the proposed dynamic graph structure and the static structure. Interestingly, our approach improves performance not only in `EvoBrain` but also in the baselines using other dynamic GNNs.

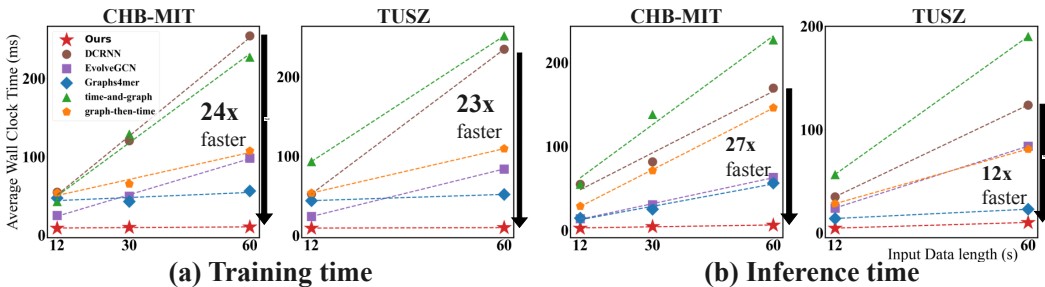

**(a) Training time**  **(b) Inference time**

Figure 4:  (a) Training time and (b) inference time vs. input data length on CHB-MIT and TUSZ datasets. Our model achieves up to **24x faster** training times and **27x faster** inference times than its competitors, demonstrating significant scalability improvements.

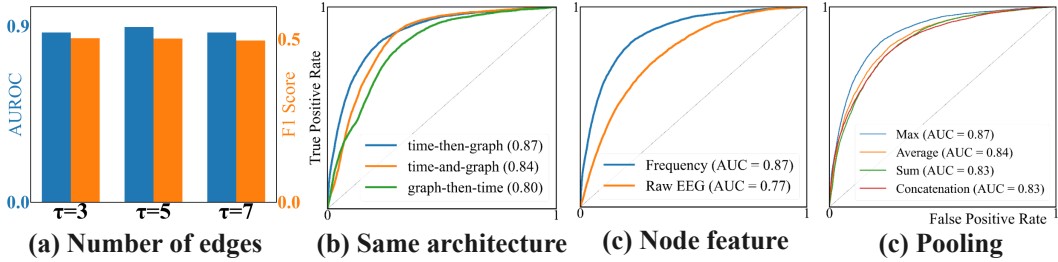

**(a) Number of edges**   **(b) Same architecture**   **(c) Node feature**   **(c) Pooling**

Figure 5: (a) Number of edges $\tau$ evaluation. (b) Approach evaluation using same architecture. (c) Results using raw EEG instead of frequency-domain features. (d) Results after modifying the node pooling mechanism.

**Computational efficiency.** To assess the computational efficiency of our method, we measured the training time and inference time. In addition to the baselines, we included time-and-graph and graph-then-time models with the same architecture as ours for comparison. dynamic GNNs require computation time proportional to the length of the input data (details are provided in Appendix C). Figure 5 (a) illustrates the average training time per step for dynamic GNNs with a batch size of 1 across various input lengths. In practice, while the RNN component operates very quickly, the GNN processing accounts for most of the computation time. Since our method performs GNN processing only once for each data sample, it is up to $24\times$ faster training time and more than $27\times$ faster inference time than DCRNN. Thus, our approach is not only superior in performance but also the fastest in terms of computational efficiency.

**Ablation study.** We set $\tau = 3$ and the top-3 neighbors' edges were kept for each node. Figure 5 (a) shows results with varying $\tau$, indicating minimal changes. This suggests that specific edges may have a significant impact. Figure 5 (b) shows the results of different approaches applied to the same architecture. Consistent with the conclusion of Theorem 1, the time-then-graph approach achieved the best performance. Figure 5 (c) shows the results when the FFT processing was removed, and raw EEG data was used as node features. The use of raw EEG data resulted in a decrease in AUROC, highlighting the importance of utilizing frequency-domain features. Figure 5 (d) presents the results when changing the pooling methods for graph classification. While we use max pooling by default, we also tested average pooling, summation pooling, and concatenation of all node features. Max pooling yielded the best performance. This suggests that seizures may occur in specific regions of the brain, allowing the model to effectively leverage the influence of particular nodes.

## 5.3  CLINICAL ANALYSIS.

We show an analysis of our constructed dynamic graphs from a neuroscience perspective. For the sake of blind review, the names are anonymized, but we **conducted this analysis with two professors who are neurosurgeons.** Figure 6 displays the top 10 edges with the strongest connections in the learned dynamic graph, where the thickness of the color represents the strength of the con-

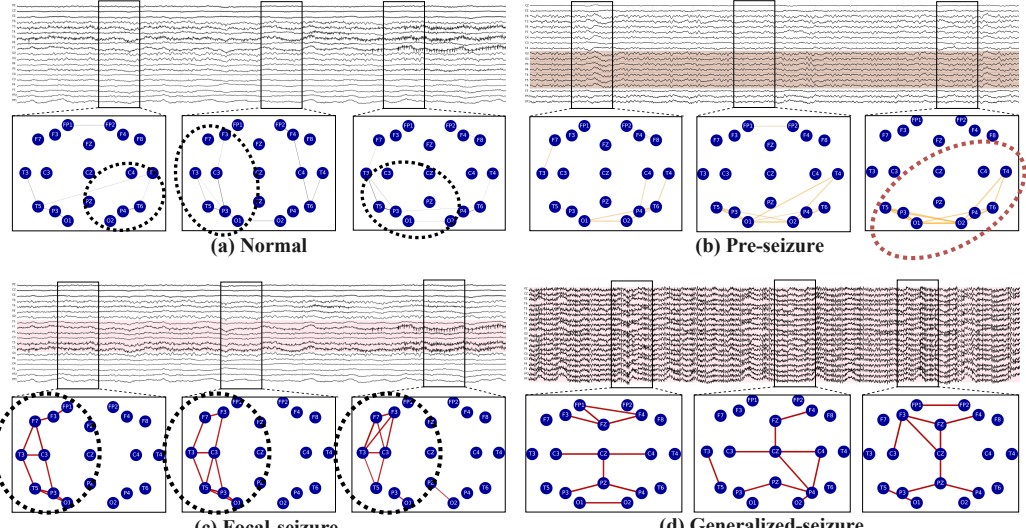

Figure 6: Learned graph structure visualizations. The color intensity of the edges indicates the strength of the connections. In (a) Normal state, the light color shows weak connections. In (b) Pre-seizure state, the connections in specific regions strengthen over time. In (c) Focal seizure, which occurs only in a specific area of the brain, strong connections are consistently present in a particular region. In (d) Generalized seizure, strong connections are observed across the entire brain.

nections. These edges are selected based on the highest $h^{edge}$ value, indicating the most significant relationships captured by the model. In Figure 6 (a), a sample unrelated to a seizure shows weak, sparse connections spread across various regions over an extended period. Figure 6 (b) shows a pre-seizure sample, where the connections between [T5, P3, O1, O2, P4, P6] gradually strengthen. This could indicate a precursor state signaling an imminent seizure. Figures 6 (c) and (d) display seizure samples, where the edges are notably stronger than in the normal state. In (c), we show the result of a focal seizure, a type of seizure that originates in a specific area of the brain, with sustained strong connections only in specific regions such as [F3, F7, P3, C3, C5, T3, O1]. In Figure 6 (d), a generalized seizure is illustrated, characterized by strong connections across the entire brain. Successful surgical and neuromodulatory treatments critically depend on accurate localization of the seizure onset zone (SOZ) (Li et al., 2021b). Even the most experienced clinicians are challenged because there is no clinically validated biomarker of SOZ. Prior studies studies have shown that abnormal connections across several channels may constitute a more effective marker of the SOZ (Scharfman, 2007; Burns et al., 2014b; Li et al., 2018). Our dynamic graph structures aligns with neuroscientific observations, successfully visualizing these abnormal connections and their changes. This offers promising potential for application in surgical planning and treatment strategies. Existing methods predominantly employed static graphs (Ho & Armanfard, 2023; Tang et al., 2022), which are unable to capture such dynamic graph structures.

## 6 CONCLUSION

In this work, we introduced a novel dynamic multichannel EEG modeling approach, `EvoBrain`, designed to address key limitations in existing seizure detection and prediction methods. By adopting a *time-then-graph* strategy, our model effectively captures the evolving nature of brain networks during seizures, providing significant improvements in both AUROC and F1 scores compared to state-of-the-art methods. Our theoretical analysis further demonstrated the expressivity advantage of *time-then-graph* over traditional approaches, and we showcased the value of dynamic graph structures in better reflecting the transient changes in brain connectivity. Looking ahead, there are several promising directions for future work. We aim to investigate explainability for both clinical applications and model transparency.

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

# Appendix

CONTENTS

## A GRAPH ISOMORPHISM AND 1-WL TEST

**Graph isomorphism** refers to the problem of determining whether two graphs are structurally identical, meaning there exists a one-to-one correspondence between their nodes and edges. This is a crucial challenge in graph classification tasks, where the goal is to assign labels to entire graphs based on their structures. A model that can effectively differentiate non-isomorphic graphs is said to have high expressiveness, which is essential for accurate classification. In many cases, graph classification models like GNNs rely on graph isomorphism tests to ensure that structurally distinct graphs receive different embeddings, which improves the model's ability to correctly classify graphs.

**1-Weisfeiler-Lehman (1-WL) test** is a widely used graph isomorphism test that forms the foundation of many GNNs. In the 1-WL framework, each node's representation is iteratively updated by aggregating information from its neighboring nodes, followed by a hashing process to capture the structural patterns of the graph. GNNs leveraging this concept, such as Graph Convolutional Networks (GCNs) and Graph Attention Networks (GATs), essentially perform a similar neighborhood aggregation, making them as expressive as the 1-WL test in distinguishing non-isomorphic graphs (Xu et al., 2019). Modern GNN architectures adhere to this paradigm, making the 1-WL a standard baseline for GNN expressivity. In our work, we also use 1-WL-based GNNs, leveraging their proven expressiveness for dynamic brain graph modeling.

## B PROOFS OF EXPRESSIVITY ANALYSIS

### B.1 EXPRESSIVENESS WITH NODE AND EDGE REPRESENTATIONS

**Lemma 2.** *[Expressiveness with Node and Edge Representations] When both node and edge representations are incorporated, a GNN can uniquely distinguish any pair of temporal EEG graphs that differ in either node features or edge features at any time step, provided the GNN is sufficiently expressive (e.g., 1-WL GNN). Details of Lemma 2 can be found in Appendix B.1.*

*Proof.* Given $\mathcal{G}^{(1)} = (\mathcal{A}^{(1)}, \mathcal{X}^{(1)})$ and $\mathcal{G}^{(2)} = (\mathcal{A}^{(2)}, \mathcal{X}^{(2)})$, suppose they differ in at least one node feature or edge feature at some time step $t$. An expressive GNN can produce different embeddings for these graphs by capturing the differences in node and/or edge features. Specifically:

1. If $\mathcal{X}^{(1)}_{:,t} \neq \mathcal{X}^{(2)}_{:,t}$ for some $t$, then the node embeddings $h^{(1)}_{i,t}$ and $h^{(2)}_{i,t}$ will differ for at least one node $v_i$.

2. If $\mathcal{A}^{(1)}_{i,j,t} \neq \mathcal{A}^{(2)}_{i,j,t}$ for some edge $(v_i, v_j)$ and some $t$, then the edge embeddings $h^{(1)}_{ij,t}$ and $h^{(2)}_{ij,t}$ will differ for that edge.

Since the GNN aggregates information from both node and edge embeddings, any difference in either will propagate through the network, resulting in distinct final representations $\mathbf{Z}^{(1)}$ and $\mathbf{Z}^{(2)}$. Thus, the GNN can uniquely distinguish between $\mathcal{G}^{(1)}$ and $\mathcal{G}^{(2)}$. □

### B.2 *time-and-graph* AND *time-then-graph*

**Lemma 4.** *[time-and-graph $\not\supseteq$ time-then-graph ] time-then-graph is strictly more expressive than time-and-graph representation family on $\mathbb{T}_{n,T,\theta}$, as time-then-graph outputs different representations, while time-and-graph does not.*

Gao & Ribeiro (2022) prove that a *time-then-graph* representation that outputs the same embeddings as an arbitrary *time-and-graph* representation. Thus, *time-then-graph* is as expressive as *time-and-graph*. To prove Lemma 4 we also provide a EEG graph classification task where any *time-and-graph* representation will fail while a *time-then-graph* would work, which then, added to the previous result, proves that *time-then-graph* is strictly more expressive than *time-and-graph*.

*Proof.* We now propose a synthetic EEG task, whose temporal graph is illustrated in Figure 7. The goal is to differentiate the topologies between two 2-step temporal graphs. Each snapshot is a static

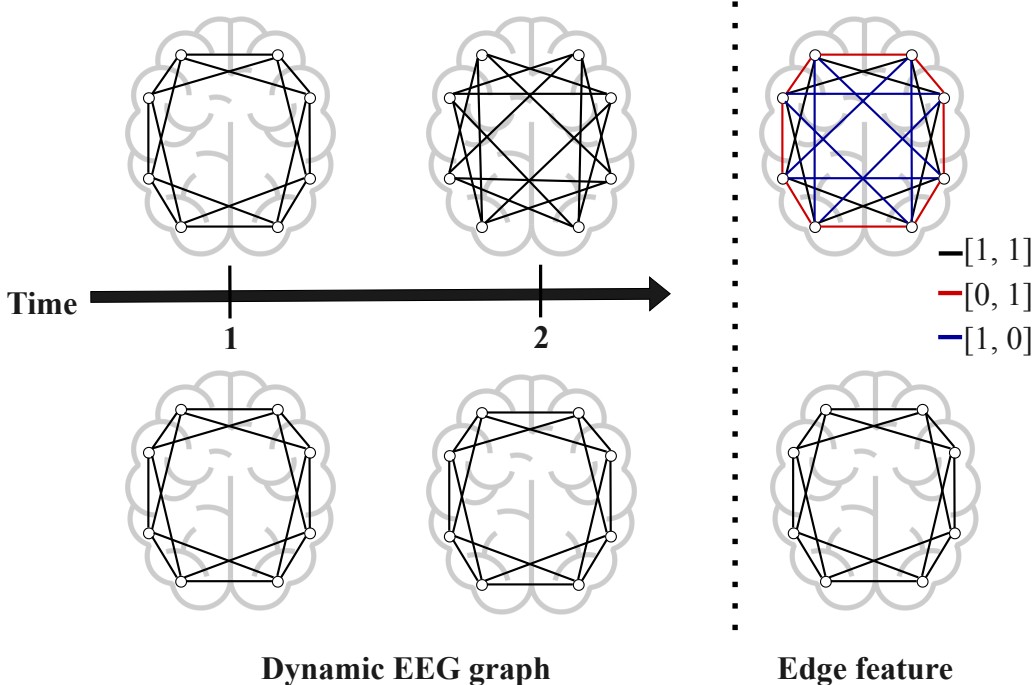

Figure 7: **A synthetic EEG task where only *time-then-graph* is expressive.** The top and bottom 2-time temporal graphs on the left side has snapshots of different structure at time $t_2$ (denote by $\mathcal{C}_{7,2}$ and $\mathcal{C}_{7,1}$). The top and bottom temporal graphs on the left show different dynamic-graph structures. The right side shows their aggregated versions, where edge attributes indicate whether they existed (1) or not (0) over time, using different colors. The goal is to distinguish the structural differences between the top and bottom graphs. *time-and-graph* have the same node representation neighbors in both snapshots, indistinguishable. *time-then-graph* aggregate the dynamic graphs into different node representations and succeeds in distinguishing them.

EEG graph with 7 attributed nodes, denoted as $\mathcal{C}_{7,s}$, where $s$ represents the smallest number of nodes on the outer circle between two neighbors which are not connected by the outer circle.

Two temporal graphs differ in their second time step $t_2$. If the graphs have the same features, any 1-WL GNN will output the same representations for both $\mathcal{C}_{7,1}$ and $\mathcal{C}_{7,2}$. We use $\mathbf{A}^{(\text{top})}$ to represent the adjacency matrix of dynamics in the top left of Figure 7, and $\mathbf{A}^{(\text{btm})}$ for dynamics in the bottom left of Figure 7. Note that $\mathbf{X}^{(\text{top})} = \mathbf{X}^{(\text{btm})}$ since the temporal graph has the same features.

Hence, for a *time-and-graph* representation,

$$\text{GNN}_{\text{in}}^L\big(\mathbf{X}_{:,1}^{(\text{top})}, \mathbf{A}_{:,:,1}^{(\text{top})}\big) = \text{GNN}_{\text{in}}^L\big(\mathbf{X}_{:,2}^{(\text{top})}, \mathbf{A}_{:,:,2}^{(\text{top})}\big) = \text{GNN}_{\text{in}}^L\big(\mathbf{X}_{:,1}^{(\text{btm})}, \mathbf{A}_{:,:,1}^{(\text{btm})}\big) = \text{GNN}_{\text{in}}^L\big(\mathbf{X}_{:,2}^{(\text{btm})}, \mathbf{A}_{:,:,2}^{(\text{btm})}\big),$$

$$\text{GNN}_{\text{rc}}^L\big(\mathbf{H}_{:,0}^{(\text{top})}, \mathbf{A}_{:,:,1}^{(\text{top})}\big) = \text{GNN}_{\text{rc}}^L\big(\mathbf{H}_{:,0}^{(\text{btm})}, \mathbf{A}_{:,:,1}^{(\text{btm})}\big), \quad \text{GNN}_{\text{rc}}^L\big(\mathbf{H}_{:,1}^{(\text{top})}, \mathbf{A}_{:,:,2}^{(\text{top})}\big) = \text{GNN}_{\text{rc}}^L\big(\mathbf{H}_{:,1}^{(\text{btm})}, \mathbf{A}_{:,:,2}^{(\text{btm})}\big).$$

Then, when we apply Equation (2) at the first time step, we get:

For the top graph:

$$\mathbf{H}_{i,1}^{(\text{top})} = \text{Cell}\bigg( \Big[\text{GNN}_{\text{in}}^L\big(\mathbf{X}_{:,1}^{(\text{top})}, \mathbf{A}_{:,:,1}^{(\text{top})}\big)\Big]_i, \Big[\text{GNN}_{\text{rc}}^L\big(\mathbf{H}_{:,0}^{(\text{top})}, \mathbf{A}_{:,:,1}^{(\text{top})}\big)\Big]_i \bigg)$$

For the bottom graph:

$$\mathbf{H}_{i,1}^{(\text{btm})} = \text{Cell}\bigg( \Big[\text{GNN}_{\text{in}}^L\big(\mathbf{X}_{:,1}^{(\text{btm})}, \mathbf{A}_{:,:,1}^{(\text{btm})}\big)\Big]_i, \Big[\text{GNN}_{\text{rc}}^L\big(\mathbf{H}_{:,0}^{(\text{btm})}, \mathbf{A}_{:,:,1}^{(\text{btm})}\big)\Big]_i \bigg)$$

Since $\mathbf{X}^{(\text{top})} = \mathbf{X}^{(\text{btm})}$, $\mathbf{A}_{:,:,1}^{(\text{top})} = \mathbf{A}_{:,:,1}^{(\text{btm})}$, and $\mathbf{H}_{:,0}^{(\text{top})} = \mathbf{H}_{:,0}^{(\text{btm})}$, we have: $\mathbf{H}_{i,1}^{(\text{top})} = \mathbf{H}_{i,1}^{(\text{btm})}$

For the second time step:

$$\mathbf{H}_{i,2}^{(\text{top})} = \text{Cell}\bigg( \Big[ \text{GNN}_{\text{in}}^L\big(\mathbf{X}_{:,2}^{(\text{top})}, \mathbf{A}_{:,:,2}^{(\text{top})}\big) \Big]_i, \Big[ \text{GNN}_{\text{rc}}^L\big(\mathbf{H}_{:,1}^{(\text{top})}, \mathbf{A}_{:,:,2}^{(\text{top})}\big) \Big]_i \bigg)$$

$$\mathbf{H}_{i,2}^{(\text{btm})} = \text{Cell}\bigg( \Big[ \text{GNN}_{\text{in}}^L\big(\mathbf{X}_{:,2}^{(\text{btm})}, \mathbf{A}_{:,:,2}^{(\text{btm})}\big) \Big]_i, \Big[ \text{GNN}_{\text{rc}}^L\big(\mathbf{H}_{:,1}^{(\text{btm})}, \mathbf{A}_{:,:,2}^{(\text{btm})}\big) \Big]_i \bigg)$$

Despite $\mathbf{A}_{:,:,2}^{(\text{top})} \neq \mathbf{A}_{:,:,2}^{(\text{btm})}$, the 1-WL GNN will output the same representations $\mathbf{H}_{i,2}^{(\text{top})} = \mathbf{H}_{i,2}^{(\text{btm})}$ for both $\mathcal{C}_{7,1}$ and $\mathcal{C}_{7,2}$. Therefore:

$$\boldsymbol{Z}^{(\text{top})} = \mathbf{H}_{i,2}^{(\text{top})} = \mathbf{H}_{i,2}^{(\text{btm})} = \boldsymbol{Z}^{(\text{btm})}$$

**Thus, Time-and-graph will output the same final representation $\boldsymbol{Z}^{(\text{top})} = \boldsymbol{Z}^{(\text{btm})}$ for two different temporal graphs in Figure 7.**

For the time-then-graph representation, we apply Equation (3):

First, for the node representations:

$$\mathbf{H}_i^{\text{node(top)}} = \text{RNN}^{\text{node}}\big(\mathbf{X}_{i,\leq 2}^{(\text{top})}\big)$$
$$= \text{RNN}^{\text{node}}\big([\mathbf{X}_{i,1}^{(\text{top})}, \mathbf{X}_{i,2}^{(\text{top})}]\big)$$
$$\mathbf{H}_i^{\text{node(btm)}} = \text{RNN}^{\text{node}}\big(\mathbf{X}_{i,\leq 2}^{(\text{btm})}\big)$$
$$= \text{RNN}^{\text{node}}\big([\mathbf{X}_{i,1}^{(\text{btm})}, \mathbf{X}_{i,2}^{(\text{btm})}]\big)$$

Since $\mathbf{X}^{(\text{top})} = \mathbf{X}^{(\text{btm})}$, we have $\mathbf{H}_i^{\text{node(top)}} = \mathbf{H}_i^{\text{node(btm)}}$ for all nodes $i$.

Now, for the edge representations:

$$\mathbf{H}_{i,j}^{\text{edge(top)}} = \text{RNN}^{\text{edge}}\big(\mathbf{A}_{i,j,\leq 2}^{(\text{top})}\big)$$
$$= \text{RNN}^{\text{edge}}\big([\mathbf{A}_{i,j,1}^{(\text{top})}, \mathbf{A}_{i,j,2}^{(\text{top})}]\big)$$
$$\mathbf{H}_{i,j}^{\text{edge(btm)}} = \text{RNN}^{\text{edge}}\big(\mathbf{A}_{i,j,\leq 2}^{(\text{btm})}\big)$$
$$= \text{RNN}^{\text{edge}}\big([\mathbf{A}_{i,j,1}^{(\text{btm})}, \mathbf{A}_{i,j,2}^{(\text{btm})}]\big)$$

Here, $\mathbf{A}_{i,j,\leq 2}^{(\text{top})} \neq \mathbf{A}_{i,j,\leq 2}^{(\text{btm})}$ for some $(i,j)$ pairs, because the graph structures differ at $t_2$. Therefore, $\mathbf{H}_{i,j}^{\text{edge(top)}} \neq \mathbf{H}_{i,j}^{\text{edge(btm)}}$ for these pairs.

Finally, we apply the GNN:

$$\boldsymbol{Z}^{(\text{top})} = \text{GNN}^L\big(\mathbf{H}^{\text{node(top)}}, \mathbf{H}^{\text{edge(top)}}\big)$$
$$\boldsymbol{Z}^{(\text{btm})} = \text{GNN}^L\big(\mathbf{H}^{\text{node(btm)}}, \mathbf{H}^{\text{edge(btm)}}\big)$$

Since $\mathbf{H}^{\text{edge(top)}} \neq \mathbf{H}^{\text{edge(btm)}}$, and 1-WL GNNs can distinguish graphs with different edge attributes, we have:

$$\boldsymbol{Z}^{(\text{top})} \neq \boldsymbol{Z}^{(\text{btm})} \tag{9}$$

**Thus, time-then-graph outputs different final representations $\boldsymbol{Z}^{(\text{top})} \neq \boldsymbol{Z}^{(\text{btm})}$ for the two temporal graphs in Figure 7, successfully distinguishing them.**

Finally, we conclude:

1. The *time-then-graph* is at least as expressive as the *time-and-graph*;

2. The *time-then-graph* can distinguish temporal graphs not distinguishable by *time-and-graph*.

Thus, *time-then-graph* is strictly more expressive than *time-and-graph*. More precisely,

$$\textit{time-and-graph} \not\succeq_{\mathbb{T}_{n,T,\theta}} \textit{time-then-graph},$$

concluding our proof.

$\square$

## C  COMPUTATIONAL COMPLEXITY ANALYSIS

In this section, we analyze the computational complexities of the three approaches: *Graph-then-time*, *Time-and-graph*, and *Time-then-graph*. We demonstrate that the *Time-then-graph* approach has the lowest computational complexity among them.

Let $T$ be the number of time steps, $V$ be the number of nodes, $E_t$ be the number of edges at time $t$, $\sum_t E_t$ be the total number of edges across all time steps, $E_{\text{agg}}$ be the number of edges in the aggregated graph (i.e., the union of all edges across time steps), and $d$ be the dimension of the node and edge representations.

### C.1  GRAPH-THEN-TIME APPROACH

In the *Graph-then-time* approach, at each time step $t$, a GNN is applied to the snapshot graph $(\mathbf{X}_{:,t}, \mathbf{A}_{:,:,t})$ to capture spatial relationships. Subsequently, an RNN processes the node embeddings over time to capture temporal dependencies.

The computational complexity per time step $t$ is dominated by:

$$\mathcal{O}\left(Vd^2 + E_t d\right),$$

where $Vd^2$ accounts for node-wise transformations (e.g., linear layers), and $E_t d$ accounts for message passing over edges.

Over all time steps, the total complexity for the GNN computations is:

$$\mathcal{O}\left(TVd^2 + \sum_{t=1}^{T} E_t d\right).$$

The RNN processes the node embeddings over time with complexity:

$$\mathcal{O}\left(VTd^2\right).$$

Therefore, the overall computational complexity of the *Graph-then-time* approach is:

$$\mathcal{O}\left(VTd^2 + \sum_{t=1}^{T} E_t d\right). \tag{10}$$

### C.2  TIME-AND-GRAPH APPROACH

In the *Time-and-graph* approach, temporal dependencies are integrated into the GNN computations. At each time step $t$, two GNNs are applied:

- $\text{GNN}_{\text{in}}^{L}$ processes the current snapshot inputs $(\mathbf{X}_{:,t}, \mathbf{A}_{:,:,t})$.

- $\text{GNN}_{\text{rc}}^L$ processes the representations from the previous time step ($\mathbf{H}_{:,t-1}, \mathbf{A}_{:,:,t}$).

The computational complexity per time step $t$ is:

$$\mathcal{O}\left(Vd^2 + E_t d^2\right),$$

due to the node-wise transformations and edge-wise message passing with updated representations.

Over all time steps, the total complexity for the GNN computations is:

$$\mathcal{O}\left(TVd^2 + \sum_{t=1}^{T} E_t d^2\right).$$

The RNN (or any recurrent unit) further processes the node embeddings with complexity:

$$\mathcal{O}\left(VTd^2\right).$$

Therefore, the overall computational complexity of the *Time-and-graph* approach is:

$$\mathcal{O}\left(VTd^2 + \sum_{t=1}^{T} E_t d^2\right). \tag{11}$$

### C.3 TIME-THEN-GRAPH APPROACH

In the *Time-then-graph* approach, temporal evolutions of node and edge attributes are modeled first using sequence models (e.g., RNNs). A GNN is then applied to the resulting static graph with aggregated temporal information.

The computational complexities are as follows:

**Node Sequence Modeling**  For each node $i \in \mathcal{V}$, an RNN processes its temporal features $\mathbf{X}_{i,\leq T}$:

$$\mathcal{O}\left(VTd^2\right).$$

**Edge Sequence Modeling**  For each edge $(i,j) \in \mathcal{E}_{\text{agg}}$, an RNN processes its temporal adjacency features $\mathbf{A}_{i,j,\leq T}$:

$$\mathcal{O}\left(E_{\text{agg}} T d^2\right).$$

**GNN over Aggregated Graph**  A GNN is applied once to the static graph with updated node and edge representations:

$$\mathcal{O}\left(Vd^2 + E_{\text{agg}} d^2\right).$$

Therefore, the overall computational complexity of the *Time-then-graph* approach is:

$$\mathcal{O}\left((V + E_{\text{agg}})\, T d^2\right). \tag{12}$$

### C.4 COMPARISON OF COMPLEXITIES

To compare the computational complexities, we consider the dominant terms in Equations equation 10, equation 11, and equation 12.

- **Graph-then-time**:

$$\mathcal{O}\left(VTd^2 + \sum_{t=1}^{T} E_t d\right).$$

- **Time-and-graph**:

$$\mathcal{O}\left(VTd^2 + \sum_{t=1}^{T} E_t d^2\right).$$

- **Time-then-graph**:

$$\mathcal{O}\left((V + E_{\mathrm{agg}}) Td^2\right).$$

By comparing these computational complexities, the **Time-then-graph** method is superior under the aggregated number of edges $E_{\mathrm{agg}}$ is smaller than the total sum of edges over all time steps, i.e., $E_{\mathrm{agg}} \ll \sum_{t=1}^{T} E_t$.

## D  IMPLEMENTATION AND MODEL TRAINING

**Data augmentation.**  During the training process, we applied the following data augmentation techniques, following prior studies (Tang et al., 2022; Eldele et al., 2021): randomly scaling the amplitude of the raw EEG signals by a factor between 0.8 and 1.2.

**Implementation details.**  We used binary cross-entropy as the loss function to train all models. The models were trained for 100 epochs with an initial learning rate of 1e-4. To enhance efficiency and sparsity, we set $\tau = 3$ and the top-3 neighbors' edges were kept for each node. The dropout probability was 0 (i.e., no dropout). `EvoBrain` has two GRUs consisting of two stacked layers and two-layer GCN with 64 hidden units, resulting in 114,794 trainable parameters. We set Our anonymous GitHub repositry (`https://anonymous.4open.science/r/EvoBrain-FBC5`) includes the source code of our `EvoBrain` and all baselines.

**Implementation of baselines.**  For baselines, DCRNN (Tang et al., 2022), EvolveGCN (Pareja et al., 2020), and LSTM (Hochreiter & Schmidhuber, 1997), we used the number of RNN and GNN layers and hidden units in our `EvoBrain`. For BIOT, we use the same model architecture described in Yang et al. (2023a), i.e., four Transformer layers with eight attention heads and 256-dimensional embedding. For CNN-LSTM, we use the same model architecture described in Ahmedt-Aristizabal et al. (2020), i.e., two stacked convolutional layers (32 $3 \times 3$ kernels), one max-pooling layer ($2 \times 2$), one fully-connected layer (output neuron = 512), two stacked LSTM layers (hidden size = 128), and one fully connected layer. Table 2 shows a comparison of trainable parameters, with our `EvoBrain` achieving the best performance using the fewest parameters.

Table 2: Comparison of trainable parameters.

|  | EvoBrain | DCRNN | EvolveGCN | BIOT | CNN-LSTM | LSTM |
|---|---|---|---|---|---|---|
| Trainable Parameters | 114,794 | 280,769 | 200,301 | 3,187,201 | 5,976,033 | 536,641 |

## E  PARAMETER SENSITIVITY OF BIOT.

Since parameter size of BIOT is larger than those of other baselines, we conducted parameter sensitivity experiments on the number of layers and embedding dimensions using TUSZ 12 seconds dataset.

| Configuration | Parameters | AUROC | F1 |
|---|---|---|---|
| Layer 4 | 3,187,201 | 0.725 | 0.325 |
| Layer 3 | 2,385,409 | 0.744 | 0.332 |
| Layer 2 | 1,596,417 | 0.708 | 0.313 |
| Embedding 256 | 3,187,201 | 0.725 | 0.325 |
| Embedding 128 | 800,769 | 0.673 | 0.261 |
| Embedding 64 | 203,777 | 0.738 | 0.341 |

Table 3: Parameter sensitivity evaluations of BIOT on layers and embedding dimensions

## F  SEIZURE PREDICTION TASK

The seizure prediction task is defined as a classification problem between inter-ictal (normal) and pre-ictal states (Burrello et al., 2020). However, there is no clear clinical definition regarding its onset or duration of preictal state (Lopes et al., 2023). So it is define as a fixed duration before the seizure occurrence (Batista et al., 2024). This duration is chosen to account for the time required for stimulation by implanted devices (Cook et al., 2013) and to allow for seizure preparation. In this study, we define the pre-ictal state as one minute, providing adequate time for effective electrical stimulation to mitigate seizures or minimal preparation. A five-minute buffer zone around the boundary between normal and seizure data was excluded from the analysis. Data labeled as seizures were discarded, and a five-minute buffer zone around the boundary data was excluded from the analysis. The remaining data were used as the normal state.

## G  DATA DESCRIPTION

Table 4: Number of EEG data samples and patients in the train, validation, and test sets on TUSZ dataset. Train, validation, and test sets consist of distinct patients.

| Task | EEG Input Length (Secs) | Train Set | | Validation Set | | Test Set | |
|---|---|---|---|---|---|---|---|
| | | EEG samples % (Pre-) Seizure | Patients | EEG samples % (Pre-) Seizure | Patients | EEG samples % (Pre-) Seizure | Patients |
| Seizure Detection | 60-s | 38,613 (9.3%) | 530 | 5,503 (11.4%) | 61 | 8,848 (14.7%) | 45 |
| | 12-s | 196,646 (6.9%) | 531 | 28,057 (8.7%) | 61 | 44,959 (10.9%) | 45 |
| Seizure Prediction | 60-s | 7,550 (9.9%) | 530 | 999 (12.0%) | 61 | 1,277 (24.4%) | 45 |
| | 12-s | 40,716 (12.8%) | 531 | 5,439 (16.0%) | 61 | 6,956 (27.6%) | 45 |

