# OpenReview forum: "Dynamic multi-channel EEG graph modeling for time-evolving brain network"
_ICLR.cc/2025/Conference — Submitted to ICLR 2025_

### Official Review · Reviewer_j4He · 2024-10-19

**Soundness:** 3
**Presentation:** 2
**Contribution:** 2
**Rating:** 5
**Confidence:** 4

**Summary:**

This paper proposes a time-then-graph model architecture called EvoBrain to capture dynamic graph structures in the brain network for EEG seizure detection and prediction. The authors show theoretical proof that time-then-graph architecture is more expressive compared to time-and-graph and graph-then-time architectures. In addition, experimental results show improved performance on seizure detection and prediction compared to several existing models.

**Strengths:**

* The authors show theoretical proof of expressiveness of three different dynamic GNNs for EEG modeling.

* Experimental results suggest that the proposed EvoBrain improves over existing GNNs on seizure detection and prediction

**Weaknesses:**

* The following paper Tang et al. 2023 also constructs dynamic graphs in a time-then-graph fashion, but there is no comparison between EvoBrain and Tang et al. 2023. Also, in related work, there is no mentioning of prior time-then-graph approaches.
  * Tang, S., Dunnmon, J. A., Liangqiong, Q., Saab, K. K., Baykaner, T., Lee-Messer, C., & Rubin, D. L. Modeling Multivariate Biosignals With Graph Neural Networks and Structured State Space Models. In Proceedings of the Conference on Health, Inference, and Learning (Vol. 209, pp. 50–71). PMLR.

* The seizure prediction task definition is unclear. On Page 7, the authors state that “we defined the one-minute period before a seizure as the preictal phase”. However, this “one-minute period” seems arbitrary. If there is preictal annotation in the dataset, please use it. If not, please provide justification on why this “one-minute period” definition is clinically meaningful.

**Questions:**

* As mentioned above, please include more time-then-graph comparisons.
* Please provide detailed descriptions on the seizure prediction task setup.
* In Figure 4, training time between different models is compared. However, for clinical utility, inference time is more crucial. Please provide comparison of inference time between the models.

---

> ### Author Response · Authors · 2024-11-25
>
> Thank you for your comments.
> We have marked the revision in the color **olive** on the paper.
>
> **W1 and Q1. Related work**.
> GRAPHS4MER (Tang et al.) differs from ours in that it simply models node features using RNNs to construct a single graph, whereas our method constructs truly dynamic graphs based on the temporal changes in frequency information. These dynamic graphs are modeled using a separate network from the node processing, allowing for a more comprehensive representation.
> Additionally, we conducted visualization experiments to demonstrate the effectiveness of our dynamic graph construction. Their work, while providing experimental results, lacks theoretical analysis, which we address in our study.
> We have updated Table 2 with new GNN baseline of GRAPH4MER.
>
>
> **W2 and Q2. seizure prediction**.
> There is no clear clinical definition regarding onset or duration of pre-ictal state.
> We define the pre-ictal state as one minute, providing adequate time for effective electrical stimulation to mitigate seizures or minimal preparation.
> We have updated this description in Appendix F.
>
>
> **Q3. Comparison**.
> We have included inference time experiments in Figure 4 (b).

---

> > ### Comment · Reviewer_j4He · 2024-11-25
> >
> > I appreciate your responses.
> >
> > 1. For Q1, please also discuss prior time-then-graph methods including GRAPHS4MER in Related Work, and clarify the key differences in your proposed method. Also, please include all GNN baseline results in Figures 2-3.
> > 2. I do not agree with the author's argument that "GRAPHS4MER differs from ours in that it simply models node features using RNNs to construct a single graph". In fact, GRAPHS4MER also constructs a graph for each EEG snapshot, which is similar to what's done in EvoBrain.

---

> > > ### Author Response · Authors · 2024-11-26
> > >
> > > We sincerely thank you for your discussion.
> > >
> > > **Q1. please also discuss prior time-then-graph methods including GRAPHS4MER in Related Work, and include all GNN baseline results in Figures 2-3**.
> > > We have included this baseline to Section 2 and updated Figure 2.
> > > Due to time constraints, we will update Figure 3 in the final version of the paper.
> > >
> > > **Q2.
> > > GRAPHS4MER also constructs a graph for each EEG snapshot, which is similar to what's done in EvoBrain**.
> > > Here, we would like to discuss this point further.
> > > In summary, we have different focuses: GRAPHS4MER employs an intermediate learning method, whereas we emphasize the role of model input.
> > > While GRAPHS4MER does learn dynamic graphs **internally** using a graph structure learning module that reconstructs graph structures for each snapshot, the input to the entire model, including the S4 model and attention layers of **Time module**, is based on the Euclidean distance or similarity of the entire data sample (e.g., 12 or 60 seconds).
> > > In contrast, our method computes the graph structure initially and directly based on individual snapshots (e.g., 1 second) from the beginning. Since Time-then-Graph is the approach to use a **Time module** (GRU) to capture graph dynamics, we aim to observe how EEG snapshots evolve over time in terms of both graph structures and node features.
> > > Therefore, it is necessary to explicitly provide a snapshot-specific graph structure to the **Time module**.
> > >
> > > Hope our clarification addresses your concern. Thank you.

---

> > > > ### Comment · Reviewer_j4He · 2024-11-26
> > > >
> > > > Thank you for your responses. I increased my rating.

---

> > > > > ### Author Response · Authors · 2024-11-30
> > > > >
> > > > > Thank you for the increased score. Regarding the remaining negative score, may we ask if you have any additional concerns? We welcome any discussion and would be happy to address them.

---

### Official Review · Reviewer_MK8A · 2024-11-03

**Soundness:** 2
**Presentation:** 3
**Contribution:** 2
**Rating:** 5
**Confidence:** 5

**Summary:**

The authors introduce a novel dynamic graph neural network (GNN) method based on the graph-then-time-based approaches for seizure detection. Specifically, a time-then-graph strategy is proposed that first models the temporal dynamics of EEG signals and graphs, and then utilizes GNNs to learn evolving spatial representations of EEG data. However, there have been related works and this work lacks novelty in the field of EEG signal processing.

**Strengths:**

The authors introduce a novel model architecture of the Dynamic Graph Neural Network method for seizure detection and superior performance has been achieved.

**Weaknesses:**

(1)	Related Works – The authors claim that “However, they do not explicitly model temporal relationships, relying instead on convolutional filters or conventional linear projections for node embeddings”. However, there have been works based on the time-then-graph-based approaches (according to Definition 3 and Lemma 3), such as Deep Feature Mining via the Attention-Based Bidirectional Long Short Term Memory Graph Convolutional Neural Network for Human Motor Imagery Recognition. The authors are encouraged to explicitly compare their approach to the specific time-then-graph methods, highlighting key differences and improvements.

(2)	Motivation – Since the time-and-graph-based approaches are more expressive, the authors are encouraged to more clearly explain the advantages of their time-then-graph approach over time-and-graph approaches, given the theoretical analysis.

(3)	Experiments and Ablation Study – The authors are advised to conduct an ablation study comparing their time-then-graph approach to graph-then-time and time-and-graph baselines using the same overall architecture and hyperparameters, to directly validate the theoretical analysis.

(4)	Results – The authors should describe how they get these results. For example, the performance of BIOT is significantly worse than the LSTM and CNN-LSTM. There would be overfitting or information leakage issues in the training process. The authors are suggested to provide more details on their experimental setup, including hyperparameters, training procedures, and any data preprocessing steps. In addition, the authors are encouraged to investigate and discuss potential reasons for BIOT's underperformance compared to simpler models.

(5)    Model Comparison – The authors are encouraged to compare their work with more latest and classic models in the field, such as EEGNet (EEGNet: A Compact Convolutional Neural Network for EEG-based Brain-Computer Interfaces), GCNs-Net (GCNs-Net: A Graph Convolutional Neural Network Approach for Decoding Time-Resolved EEG Motor Imagery Signals), Bi-LSTM-GCNNet (Deep Feature Mining via the Attention-Based Bidirectional Long Short Term Memory Graph Convolutional Neural Network for Human Motor Imagery Recognition), and EEGMamba (EEGMamba: Bidirectional State Space Model with Mixture of Experts for EEG Multi-task Classification).

**Questions:**

(1) What is the unique and significant technical contribution of your work? The technical idea of this work is quite similar to related works in the field.

(2) Why don't compare the performances of the graph-then-time, time-and-graph, and time-then-graph approaches using the same overall architecture and hyperparameters, to directly validate the theoretical analysis? The current experiments cannot support your theoretical analysis.

(3) Considering that the time-and-graph-based approach is more expressive in feature learning, why not design a novel architecture based on it?

---

> ### Author Response · Authors · 2024-11-25
>
> Thank you for your time, and
> we have marked the revision in the color **blue** on the paper.
>
> **W1. Related work**.
> BiLSTM-GCNNet (Hou et al.) differs from ours in that it simply models node features using RNNs to construct a single graph, whereas our method constructs truly dynamic graphs based on the temporal changes in frequency information. These dynamic graphs are modeled using a separate network from the node processing, allowing for a more comprehensive representation.
> Additionally, we conducted visualization experiments to demonstrate the effectiveness of our dynamic graph construction. Their work, while providing experimental results, lacks theoretical analysis, which we address in our study.
>
>
> **W2. and Q3. Motivations**.
> As stated in Theorem 1, the time-then-graph model is more expressive, rather than the time-and-graph.
>
> **W3. and Q2. Experiments and Ablation Study**.
> We conducted an ablation study comparing their time-then-graph approach to graph-then-time and time-and-graph baselines using the same overall architecture and hyperparameters, to directly validate the theoretical analysis in Figure 5 (b) and Figure 4.
>
> **W4. Results**.
> Detailed descriptions of our experimental setup, including hyperparameters, training procedures, and data preprocessing steps, are provided in Appendix D.
> Also we have conducted parameter sensitivity experiments of BIOT in Appendix E.
>
> **W5. Model Comparison**.
> We did not include EEGMamba and Bi-LSTM-GCNNet as the official code for these models is not publicly available, making fair comparison challenging.
> EEGNet and GCNs-Net, while classic models, rely on CNN and GCN-based architectures that have already been covered in our experiments.
> Instead, we have added experiments comparing our model to more recent approaches, such as GRAGHS4MER, and other models with similar architectures to ours.
> We have undated the new results to Table 2.
>
> **Q1. Contributions**.
> The study highlights the necessity of using dynamic graphs for EEG and proposes their construction. Its effectiveness is theoretically demonstrated with a new assumption involving node features and validated through experiments on real clinical data, including a novel seizure prediction task.

---

> > ### Comment · Reviewer_MK8A · 2024-11-28
> > **Feedback on the authors ' comments**
> >
> > (1) W1. Related work.
> > Thanks for the authors' feedback. The authors are encouraged to compare the performance with Hou's work (BiLSTM-GCNNet).
> >
> > (2) W2. and Q3. Motivations & W3. and Q2. Experiments and Ablation Study.
> > The authors are encouraged to implement the "graph-then-time" and "time-and-graph" baselines under the authors' proposed architectures instead of the architecture of EvolveGCN and DCRNN, for a fair comparison.
> >
> > (3) W5. Model Comparison.
> > The Bi-LSTM-GCNNet source codes should be publicly accessible, such as the codes from the EEG-DL library (https://github.com/SuperBruceJia/EEG-DL/blob/master/Models/Network/BiLSTM_with_Attention.py).
> >
> > (4) Q1. Contributions.
> > Unfortunately, the theoretical analysis is not rigorous enough to show the "time-then-graph" is more expressive than other architectures.

---

> ### Author Response · Authors · 2024-11-25
> **Question from Authors.**
>
> Considering W2-motivation and Q3, may I ask why you gave confidence 5. It seems you did not even read basic claim of our paper, Theorem 1, which concludes that the time-then-graph model is more expressive than the time-and-graph model.

---

> > ### Comment · Reviewer_MK8A · 2024-11-28
> > **Feedback to the authors**
> >
> > Unfortunately, the theoretical analysis is not rigorous enough to prove that the "time-then-graph" is more expressive than other architectures. The current "theoretical analysis" is simply some claims by the authors.

---

> ### Author Response · Authors · 2024-11-30
>
> Thank you for your feedback.
>
> (1,3) The Bi-LSTM-GCNNet source code does not include essential components for constructing the graph structure, which we believe are critical for reproducing the results described in the paper, as the graph structure and its dynamics over time are the main focus of our work.
>
> (2) As mentioned in our last response, **we referred to your comment and have already implemented** the "graph-then-time" and "time-and-graph" baselines within the framework of our proposed architecture, as shown in Figure 5 (b) and Figure 4.
> We **have already highlighted our update in blue in the caption of Figure 5 (b)**.
>
> (4) Theoretically, we arrive at the same conclusion as [1]; the proof differs and is approached from different perspectives as we focus on EEG dynamics.
> As we discussed with Reviewer-C4vK, the theoretical expressivity analysis presented in Appendix A.2 of the paper [1] provides general proof for unattributed or edge-only graphs, evaluating the model's ability to distinguish graph structures based solely on their connectivity without considering node features.
> In contrast, in the case of EEG graphs, the structure construction is based on **node features** (lines 80-83), specifically the similarity measures between features of two EEG electrodes.
> As outlined in lines 43-46, the node embeddings and similarity measures are key factors in determining the graph's construction.
> In Lemma 2 and proof in Appendix B.1, we evaluate the necessity of incorporating both node and edge information to effectively distinguish dynamic EEG graphs for modeling seizures.
> Further, we would address the overall goal and contribution of this paper. It is not yet certain if time-then-graph will prove to be an effective choice in EEG modeling, nor when it might be preferable to recent graph-then-time and time-and graph.
> However, we cannot even begin to answer the utility of this new modeling strategy until we develop an EEG seizure analysis method to use it.
> The purpose of this paper is to provide a theoretical analysis and foundational guideline on how to model temporal-graph dynamics for EEGs, which may eventually lead to more accurate models.
> As mentioned in the survey paper (arXiv:2404.18211) Reviewer-C4vK introduced, numerous studies employ various dynamics-GNN architectures.
> However, as stated in the conclusion section, "Dynamic GNNs still face challenges, such as **scalability, theoretical guidance**, and dataset construction," there are unresolved challenges, particularly regarding the theoretical aspects.
> Our work is the first to provide theoretical guidance for EEG-based seizure research.
> Again, our analysis aligns with the general conclusions presented in [1].
> However, since this is a general edge-level study and does not focus on EEGs, we have built upon their approach by proposing a dynamic graph construction method, updating theoretical analyses to node-level, and conducting experiments on real-world EEG datasets.
>
> Considering your simple claim that the theoretical analysis is not rigorous enough and that "the time-and-graph-based approach is more expressive," without further descriptions, **could you provide specific reasoning, evidence, or details regarding aspects that may be unclear or insufficiently stated?**
> Thank you.
>
> [1] Jianfei Gao and Bruno Ribeiro. On the equivalence between temporal and static equivariant graph representations. In Proceedings of the 39th International Conference on Machine Learning, pp.7052–7076, 2022

---

> ### Comment · Reviewer_MK8A · 2024-12-02
> **Feedback from the reviewer**
>
> We appreciate your detailed responses! However, this work simply combined the GRU and GCN models into a "time-then-graph" framework, which is unfortunately not technically novel enough for publication in ICLR. Some suggestions are as follows:
>
> (1) Lack of technical innovation - A promising technical innovation is required for publication in ICLR, besides the model combinations and result analysis.
>
> (2) True nature of brian dynamics - A comprehensive analysis and demonstrations of the brain dynamics are required, whereas Figure 6 (Learned graph structure visualizations.) and Figure 7 (A synthetic EEG task where only time-then-graph is expressive.) are unfortunately not enough to demonstrate the dynamics of brain networks. Meanwhile, the authors presented a directed graph whereas in Figures 6 and 7, they illustrated the dynamic graphs in an undirected way.
>
> (3) Task-agnostic experimental design - The authors are encouraged to implement their method into different EEG-related fields, such as emotion recognition, motor imagery, etc. The current two-class seizure classification task is limited to show the effectiveness of the introduced method and the current paper's title is unfortunately not suitable under only the seizure prediction task.
>
> (4) Notations - The mathematical notations in the manuscript are kind of messy, and we encourage the authors to use the IEEE math formatting.
>
> I hope these comments will be helpful.

---

> ### Author Response · Authors · 2024-12-03
>
> We thank you for your time, feedback, and the increased score.
>
> (1) As mentioned in our last response, we would like to emphasize that this work goes beyond a simple combination by providing a theoretical analysis and foundational guidelines on how to model temporal-graph dynamics for EEGs. This approach includes a novel dynamic graph construction method and may eventually lead to more accurate models.
> Indeed, a novel method or model architecture tailored for modeling EEG dynamics is highly appealing.
> We recognize that this direction has become an important line of inquiry, and we are beginning to understand how a general analysis can support broader modeling efforts.
>
> (2)(3)(4) Thank you for commenting illustrations and title.
> Due to the policy regarding PDF updates, in the final version of the paper, we will update the illustrations to directed graphs and reconsider the title.
> We understand other EEG-related fields, such as emotion recognition or motor imagery, are promising future directions.
> Also, we will review and refine the mathematical notations in the final version.
>
> Here, we thank you for your discussion.
> We are welcome to any further concerns if you have and would be happy to address them.

---

### Official Review · Reviewer_C4vK · 2024-11-03

**Soundness:** 2
**Presentation:** 3
**Contribution:** 2
**Rating:** 6
**Confidence:** 3

**Summary:**

This paper introduces a dynamic graph neural network approach called EvoBrain for seizure detection using multi-channel EEG data. EvoBrain implements a strategy called "time-then-graph" strategy, as opposed to "graph-then-time" and "time-and-graph" strategies, while providing a theoretical analysis proving the expressivity advantage of this method. Three key advantages claimed by the authors include (1) the actual inclusion of dynamic graph structures throughout time, instead of a static graph structures which the paper claims previous papers using dynamic graphs actual use, (2) consistent stronger results supported by table 2, and (3) seizure prediction at the preictal state, unlike most previous detection algorithms.

**Strengths:**

In general, this seems to me to be quite a well thought paper, well framed in the wide literature, with significance to the clinical application of seizure prediction. My knowledge in this (sub)field is limited and - as I'll highlight in the weaknesses section - there are some parts that are confusing to me; however, the paper seems to be quite strong on the three main points I mentioned in the Summary: (1) inclusion of actual dynamic graph structures throughout time, instead of a static graph structures which the paper claims previous papers actually used, (2) consistent stronger results showed in table 2, and (3) seizure prediction at the preictal state, unlike most previous detection algorithms. All of this while providing a theoretical analysis on the expressive power of several points in the paper under the 1-WL graph isomorphism test.  For all these strengths, I believe the paper's originality come from the creative combination of existing ideas to solve known limitations in prior results, apparently successfully. As a consequence, the model's ability to predict seizures 1 minute ahead of time is not only relevant to the clinical field, but also to the computational field as this is done in a computationally efficient fashion with a solid theoretical analysis.

Some other specific strengths of this work that I'd like to highlight are: (1) EvoBrain's gains in performance were achieved with a 23x faster design than the SOTA baseline, (2) validation of this model on an external dataset, (3) the evaluation of the importance of the dynamic graph structure illustrated in figure 3, which not only shows that this paper's ideas could be used to improve previous models, but also that if one wants even faster models, maybe the dynamic graph calculation might not be needed, (4) the computational efficiency analysis, and (5) the ablation analysis on the FFT preprocessing.

Based on these points, I recommend acceptance to ICLR. However, my recommendation is only marginally above the acceptance threshold because of the reasons I'll mention in the Weaknesses section, which I believe will be possible to be mostly tackled during the rebuttal period.

**Weaknesses:**

One main weakness I identify in this paper is that it's not clear to me how strong/true their claims on novelty are, specifically on two points:
1. Graph-then-time novelty. The paper gives the impression that the "time-then-graph" approach is new/introduced for the first time in this paper. This seems supported by the fact that in table 2 only EvoBrain is shown as the "time-then-graph" type. However, it seems to me that this approach was actually introduced in Gao and Ribeiro (2023), thus I ask the authors whether they can clarify how their approach differs from or build upon Gao and Ribeiro (2023), and explain why that work wasn't included as a baseline in the experiments. This would help address the novelty concern more directly.
2. Claim that only in this paper they truly use "dynamic" graphs. Between lines 67-71, the paper makes a key claim for this work's novelty, in which it is argued that previous GNN works for seizure prediction labelled as "dynamic" are actually based on static graph structures, and that only the temporal aspect of the nodes was considered dynamic before. In my opinion, there are contradictions in this paper with regards to this claim, which require further clarification over the rebuttal period. For example, in lines 52-53, it is said that in the graph-then-time approach, "learning channel correlations" are conducted "at each time step", in practice meaning that indeed the graph is dynamic because at each time step a new graph structure is used. Furthermore, in the formulations in equations 1 and 2, the adjacency matrices change at each time step, just like the "time-then-graph" strategy. Finally, figure 1 doesn't seem to make this distinction clearly either. My impression is that the "time-then-graph" strategy is not so much about actually having "dynamic" graphs, but more how all snapshots up to each time step are considered to create both the adjacency matrix and the node/edge features. I'm wondering whether my confusion also comes from the formulation in section 3.1, in which I do not understand why there is a pairwise connectivity between two channels represented as $e_{ij}$ without a time component, and how it differs from the corresponding $a_{ij}$ and $h_{ij,t}$ representations. This is such an important part of the paper, that I hope the authors can make this more clear during the rebuttal period and in the final version of the paper. To be specific and to summarise this point, I'm asking the authors to provide a more precise definition of what they mean by "dynamic" graph structures, and how their approach specifically differs from previous methods in this regard. This would help clarify their novelty claim.


A further main weakness I identify in this paper is the experiments analysis on the detection/prediction models performances, specifically on three points:
1. There is no variation measure on the experiments results, which would be particularly important in table 2. I'm guessing that the authors did not want to perform cross validation given the TUSZ dataset already came with pre-defined train-validation-test splits and they use an external dataset, but if this was the case then I suggest the authors should at least repeated their experiments multiple times with different random initialisations of the models to report mean and standard deviations in table 2. Some of the results in table 2, even though consistent, make one wonder that maybe there wouldn't be a significant difference if variation (eg standard deviation or variance) is considered. Also, such close results in figure 3 suggest that if such variation calculations were performed, it would indicate that maybe those differences wouldn't be as significant.
2. The fact that no traditional machine learning model was used. For example, what do clinicians using computational models develop, assuming a lot do not use deep learning? Some times traditional machine learning models like SVM or random forests can be as strong or stronger in a certain prediction task, so even if this model brings other advantages compared to previous deep learning models, in a conference like ICLR I believe it is important to understand how that compares to traditional ones that are easier and faster to train and run inference on. My experience comes more from fMRI data, where big simplifications can be done on the temporal data producing some very good results, and I'm guessing that is the same with EEG signal processing. Could the authors then include comparisons with traditional ML methods commonly used in clinical practice for seizure detection/prediction, such as support vector machines or random forests. This would provide a more comprehensive evaluation of the model's practical utility.
3. I surely commend the paper suggesting a "simple" GNN model like the GCN for good and fast results. However, with so many GNN models out there, one would expect at least one or two more "complex" GNN models to understand whether the performance would improve at the cost of more parameters or slower training.



These are the two main weaknesses which will be the most important ones for me to change my scoring during the rebuttal period, if properly tackled.

**Questions:**

As I mentioned, the two main weaknesses I have identified in the previous section will be the main points in which a response from the authors will most likely change the score I give to this paper. I leave though the following questions and suggestions which could help me better understand whether/how to increase the score of this paper:

1. There is a significant class imbalance highlighted in table 1 but the paper doesn't mention anything about it. Did the authors tackled it in some specific way?
2. How did the authors decide the hyperparameters to use, and did it follow the train-validation splits? For a fair comparison, how did that differ from the hyperparameter search of the other models? There is one hyperparameter that seems quite important to me (the $\tau$ threshold introduced in line 263) which was not analysed/ablated.
3. In lines 163/164 it is said "If $e_{ij}$ exists". Maybe this is connected to my confusion mentioned in the previous section, but how could this value not exist? Furthermore, it is said in the same lines that $a_{ij,t} \in R^e$; how is $e$ defined in this case?
4. The authors seemed to have forgotten to write some words in line 181 for it to make sense grammatically.
5. I do not understand why the EEG graphs are directed (as mentioned in line 268). Shouldn't graphs based on correlations be undirected?
6. I'd ask the authors to clarify what they mean as "our method performs GNN processing only once" in line 451. I think it's very clear from the paper that the message-passing component of GNNs, as well as the construction of the (dynamic) graphs at each time step is used at each epoch, so I'm not sure I understand what is this "GNN processing" that is run "only once". This surely would also help better understand the depiction in figure 1.
7. It's not clear to me how the analysis provided in section 5.3 could be generalised for useful clinical practice, even though the specific analysis in this section was validated by two neurosurgeons. Given the limited performance in the F1 metric, I'm guessing that the seizure prediction for each person needs to be much more complex and varied than just the main networks identified in figure 5. Could the authors maybe write one or two sentences on how this depiction could vary across different predictions and be used by neurosurgeons? (eg, is it only the connections getting stronger, or also some specific EEG regions?)

I finish with two specific suggestions about the code shared by the authors:
1. The `requirements.txt` file only has specific versions for 3 packages, and a range of versions for one package. For better reproducibility consider using a .yaml file generated from mamba/conda if you are using such package manager, or something else to ensure that any person could avoid reproducibility issues by starting with the specific dependencies used by the authors.
2. It's not recommended to import everything from specific modules (ie, using `import *`). Authors can see an old discussion about the topic here: https://stackoverflow.com/questions/2386714/why-is-import-bad

---

> ### Author Response · Authors · 2024-11-25
>
> We appreciate your detailed review and the recognition of the strength of our work.
> We have marked the revision in the color **magenta** on the paper.
>
> **W1-1. Novelty concern**.
> While we arrive at the same conclusion, the proof differs partially and is approached from different perspectives.
> We have updated and clarified this in the Introduction on lines 37-43 and 80-83.
> Epilepsy is fundamentally a network disease, where abnormal connections that switch from a stable (non-seizure) state to an unstable (seizure) state across multiple EEG channels may serve as a more effective marker [1] (line 37-40).
> Therefore, our goal is to capture how such channel connections (formulated as a graph) change over time (temporal dynamics).
> The theoretical expressivity analysis presented in Appendix A.2 of the original paper provides a general proof for unattributed or edge-only graphs, evaluating the model's ability to distinguish graph structures based solely on their connectivity, without considering node features.
> In contrast, in the case of EEG graphs, the structure construction is based on **node features** (line 80-83), specifically the similarity measures between features of two EEG electrodes.
> As outlined in line 41-43, the node embeddings and similarity measures are key factors in determining the graph's construction.
> In Lemma 2 and proof in Appendix B.1, we evaluate the necessity of incorporating both node and edge information to effectively distinguish dynamic EEG graphs for modeling seizures.
> Overall, this paper is the first to provide a theoretical analysis of EEG-oriented graphs that incorporate both nodes and edges.
> Our experiments highlight the necessity of using dynamic graphs as well as our theoretical finding, i.e., time-then-graph.
>
> [1] Adam Li et,al. Neural fragility as an eeg marker of the seizure onset zone. Nature Neuroscience,pp.1–10,2021.
>
> **W1-2. Dynamic graphs**.
> Thank you for your comments. We have updated Section 3.1 and 4.1.
>
> We would like to clarify the distinction between "dynamic graph structures" and "dynamic graph modeling" as follows:
> Dynamic Graph Structure: Unlike previous methods that typically construct a single static graph per data sample over all snapshots and only model temporal changes at the node level, we introduce a dynamic graph structure where the graph itself evolves over time.
> As described in Section 4.1, we propose a method to construct time-varying adjacency matrices based on frequency information, allowing the graph topology to adapt at each snapshot dynamically.
> Dynamic Graph Modeling: To fully leverage this dynamic graph structure, we emphasize the importance of the time-then-graph approach, which processes all snapshots up to each time step to construct both the adjacency matrix and the node/edge features. This is distinct from other methods where temporal and spatial aspects are handled separately. Our proposed EvoBrain builds on this principle, representing the dynamic graph with the highest expressivity.
>
> **W2-1. Variation measure**.
> We have now repeated our experiments five times with different random initializations of the models and included variation measures for 12 second detection task of TUSZ dataset.
> Due to time constraints, we will update all experiments in the final version of the paper.
>
> | Model         | Metric | Ave   | Std        |
> |---------------|--------|-------|------------|
> | **SVM**       | AUROC  | 0.765 | ±0.004     |
> |               | F1     | 0.369 | ±0.007     |
> | **RF**        | AUROC  | 0.778 | ±0.004     |
> |               | F1     | 0.354 | ±0.005     |
> | **LSTM**      | AUROC  | 0.794 | ±0.006     |
> |               | F1     | 0.381 | ±0.019     |
> | **CNN-LSTM**  | AUROC  | 0.754 | ±0.009     |
> |               | F1     | 0.354 | ±0.011     |
> | **BIOT**      | AUROC  | 0.726 | ±0.016     |
> |               | F1     | 0.320 | ±0.018     |
> | **EvolveGCN** | AUROC  | 0.757 | ±0.004     |
> |               | F1     | 0.343 | ±0.012     |
> | **DCRNN**     | AUROC  | 0.817 | ±0.008     |
> |               | F1     | 0.415 | ±0.039     |
> | **GRAPGS4MER**| AUROC  | 0.833 | ±0.005     |
> |               | F1     | 0.413 | ±0.017     |
> | **EvoBrain**  | AUROC  | 0.869 | ±0.003     |
> |               | F1     | 0.506 | ±0.009     |

---

> > ### Author Response · Authors · 2024-11-25
> >
> > **W2-2. Traditional methods**.
> > We have conducted experiments using SVM and random forests, and updated the Table 2.
> >
> > **W2-3. "Complex" GNN baseline**.
> > We have updated Table 2 with new GNN baseline of GRAPH4MER.
> >
> > **Q1. Class imbalance**.
> > Class imbalance may be beyond the primary scope of our study, so we have not made specific contributions in this area. However, to address this concern, we used AUROC and the F1 score at the optimal threshold as evaluation metrics.
> >
> >
> > **Q2. Experimental setting**.
> > We set the baseline model's parameters to the optimal values proposed in the original paper (Appendix A3).
> > For the BIOT, we conducted parameter sensitivity experiments on the number of layers and embedding dimensions.
> > For dynamic GNN models, we matched parameters such as the number of layers and dimensions using grid search.
> > Additionally, we have updated an ablation study on $\tau$ in Figure 5(a).
> > We used the official train-validation splits for all experiments.
> >
> >
> > **Q3. How is $e$ defined?**.
> > $e_{i,j,t}$ is estimated using channel metrics, which in our case is the similarity measure between EEG channels.
> > A top-$\tau$ selection is then applied to determine existence or non-existence.
> > The $a_{i,j,t}$ is then defined based on existence or non-existence of edge.
> >
> > **Q4. Grammer**.
> > Thank you, we have fixed it.
> >
> >
> > **Q5. Undirected graph**.
> > A fully connected correlation graph is initially undirected.
> > However, it becomes directed by designating a node as an anchor and limiting the number of edges per node to its top $\tau$ neighbors.
> >
> >
> > **Q6. GNN processing only once**.
> > The time-then-graph model performs GNN processing only at the final time step of EEG time snapshot for each training or inference step.
> > In contrast, the time-and-graph and Graph-then-Time models apply GNN processing at each EEG time snapshot.
> >
> > **Q7. How the analysis provided in section 5.3 could be generalised for useful clinical practice**.
> > Here we have updated this section in the revised PDF.
> > Clinically, abnormal waveforms or connections in EEGs originating from specific brain regions, i.e., seizure onset zone (SOZ), contribute significantly to seizure detection.
> > However, there is no clinically validated biomarker for SOZ.
> > We visualized the temporal evolutions of different EEG snapshots (e.g., 1 second to 3 seconds or to 10 seconds) to assess how brain connections differ during various states and how these connections consistently change over an EEG.
> > We then collaborated with neurosurgeons to evaluate whether these visualizations provide meaningful insights, such as whether the observed differences imply clinical significance or are merely variations without diagnostic value.
> >
> >
> > **Code suggestion**.
> > Thank you. We have updated it.

---

> > > ### Comment · Reviewer_C4vK · 2024-11-25
> > >
> > > I thank the authors for their detailed answer to my questions and overall answers to my criticisms. I believe most of my points were tackled and things are much clearer now. Given the late time in the rebuttal period, I'll focus on two important points that I believe were not answered so I'd like to hear from the authors if they still have time.
> > >
> > > 1. I believe the new lines 80-83 make it clearer how this work is different from Gao and Ribeiro (2023)'s work. However, the authors still did not include this previous work as a baseline in the experiments. Have I missed something here?
> > > 2. I appreciate the authors trying to make the distinction between "dynamic graph structures" and "dynamic graph modelling", but it seems to me my previous point is kept. The authors are once again saying that previous methods "typically construct a single static graph per data sample over all snapshots", and I'm arguing this is not the case. To support my point, in my original review I've mentioned some points in this paper that I believe contradict this statement (without even me needing to point to specific papers), in which indeed it seems to me that in previous methods models create a different graph per snapshot/timestep. Some of the clarifications in this rebuttal and from what I see in the updated paper seem to support that the authors arrive to a much better way to create graphs at each snapshot, rather than the bold statement that previous works do not have a different (thus dynamic) graph adjacency matrix at each snapshot.

---

> > > > ### Author Response · Authors · 2024-11-26
> > > >
> > > > We sincerely thank you for your thoughtful feedback.
> > > >
> > > > **Q1. The authors still did not include Gao and Ribeiro (2023)'s work as a baseline in the experiments**.
> > > > Architecturally, we used the same GRU-GCN backbone as in Gao and Ribeiro (2023)'s work, so we cannot include this to the baselines.
> > > >
> > > > **Q2. Dynamic graph**.
> > > > Thank you, we would further clarify step-by-step for this question.
> > > > As described in Section 3.1, we use node feature $\mathcal{X}\in\mathbb{R}^{d\times N\times T}$ and  adjacency matrix $\mathcal{A}\in\mathbb{R}^{N\times N\times T}$, where N is number of nodes, d is node feature dimention, and T is snapshot length.
> > > >
> > > > **2-1. Between lines 67-71 (new lines 82-87), the paper makes a key claim for this work's novelty, in which it is argued that previous GNN works for seizure prediction labelled as "dynamic" are actually based on static graph structures, and that only the temporal aspect of the nodes was considered dynamic before**.
> > > > Previous studies use node feature $\mathcal{X}\in\mathbb{R}^{d\times N\times T}$ (same to ours) and  adjacency matrix $A\in\mathbb{R}^{N\times N}$, which lacks time dimention.
> > > > In such cases, while the node features $\mathcal{X}$ vary across $T$ snapshots, the same adjacency matrix $A$ is applied uniformly across all time snapshots.
> > > > As a result, all time steps or snapshots share the same graph structure during time learning, and no graphically temporal or dynamic dependencies are captured.
> > > >
> > > > **2-2. For example, in lines 52-53 (new lines 53-54), it is said that in the graph-then-time approach, "learning channel correlations" are conducted "at each time step", in practice meaning that indeed the graph is dynamic because at each time step a new graph structure is used**.
> > > > For existing works, throughout the graph-then-time process, only the node representations $H^\text{node}\in\mathbb{R}^{d'\times N\times T}$ are computed iteratively at each timestep.
> > > > Edge features remain shared across all timesteps because the adjacency matrix $A \in \mathbb{R}^{N \times N}$ is fixed.
> > > > Even if the graph-then-time approach employs a sequence of GNNs for each timestep, all updated node embeddings are derived from shared graph structures then for temporal modeling (e.g., in RNN-based models).
> > > > Indeed, for a truly dynamic approach, it is necessary to compute the adjacency matrix $A$ for different timesteps and evaluate how both the graph structures and node features evolve over time.
> > > > This is our proposal in Section 4.1.
> > > >
> > > > **2-3. Furthermore, in the formulations in equations 1 and 2, the adjacency matrices change at each time step, just like the "time-then-graph" strategy**.
> > > > Yes, all three modeling approach, including those outlined in equations 1 and 2, has the capacity to handle dynamic graphs where we propose constructing the adjacency matrices change at each time step ($\mathcal{A}\in\mathbb{R}^{N\times N\times T}$, rather than $A \in \mathbb{R}^{N \times N}$).
> > > > Our experiment in Figure 3 shows that such new adjacency matrices $\mathcal{A}$ are fundamental for general time and graph modeling strategies.
> > > >
> > > > **2-4. Figure 1 doesn't seem to make this distinction clearly either**.
> > > > We will continue revising Figure 1 until the author deadline on 11/27 AOE.

---

> > > > > ### Comment · Reviewer_C4vK · 2024-11-27
> > > > >
> > > > > I really appreciate the authors' time to answer my questions.
> > > > >
> > > > > I'm finding it difficult to understand the authors' claims that they have differences from Gao and Ribeiro (2023)'s work, but at the same time it seems at its essence this work has the same backbone, thus I'm finding difficult to understand how this work is novel from an architectural perspective, which I believe is a claim of this paper. If there is an architectural difference, then it should be possible to use Gao and Ribeiro's work as baseline but, if there is not, then there's no novelty from an architectural perspective? If the authors base their motivation on previous works not related to EEG, then why did the authors focus on this very specific paper when there are so many papers on (truly) dynamic GNNs (see for example surveys arXiv:2405.00476 and arXiv:2404.18211)?
> > > > >
> > > > > Furthermore, on the claim that this is the first paper to use truly dynamic graphs on EEG, the claims used by the authors in this rebuttal process do not seem to match the notation used in equations 1 and 2, where it's possible to see that the adjacency matrices can vary across time (ie, they have a "t" component). If indeed this is not the case, maybe a clarification should be added/modified to the paper to make this clearer.
> > > > > (Finally, I must admit in the process of this rebuttal I've found this one paper (10.1038/s41598-022-23656-1) which I believe uses dynamically-generated graphs in a similar fashion as claimed by this paper - the method is fundamentally different so I believe it doesn't change much in this review, but I must mention it as the one example I found)

---

> > > > > > ### Author Response · Authors · 2024-11-27
> > > > > >
> > > > > > We sincerely thank you for your feedback and welcome to further discussion.
> > > > > >
> > > > > > **Q1. I'm finding difficult to understand how this work is novel from an architectural perspective. Why did the authors focus on Gao and Ribeiro's work?**
> > > > > > From architectural perspective, we use the same GRU-GCN for EEGs as Gao and Ribeiro's general GNN work, but with a similar methodological approach to Tang et al. [1], which employs a already proposed DCRNN as the backbone.
> > > > > > We are not saying this is acceptable, we would like to address the overall goal and contribution of the paper.
> > > > > > It is not yet certain if time-then-graph will prove to be an effective choice in EEG modeling, nor when it might be preferable to recent graph-then-time and time-and graph.
> > > > > > We cannot even begin to answer the utility of this new modeling strategy until we develop EEG seizure analysis method to use it.
> > > > > > The purpose of this paper is to provide a theoretical analysis and foundational guideline how to modeling temporal-graph dynamics for EEGs, which may eventually lead to more accurate models.
> > > > > > As mentioned in the survey papers you introduced, there are numerous studies employing various architectures. However, as stated in the conclusion section of arXiv:2404.18211, "Dynamic GNNs still face challenges, such as scalability, **theoretical guidance**, and dataset construction," there are unresolved challenges, particularly regarding the theoretical aspects.
> > > > > > In this context, the Gao and Ribeiro's work is highly novel as it provides general theoretical contributions.
> > > > > > Additionally, this work partially addresses another challenge, scalability, which is also highlighted in the surveys. However, since this is a general study and does not focus on EEGs, we have built upon their approach by proposing dynamic graph construction method, updating theoretical analyses to node-level (has stated in our previous responses), and conducting experiments on real-world EEG datasets.
> > > > > > Indeed, a novel method or model architecture tailored for modeling EEG dynamics is highly appealing.
> > > > > > We recognize this direction has become an important line of inquiry and we are beginning to understand how a general analysis can support broader modeling efforts.
> > > > > >
> > > > > > [1] Siyi Tang et al., Self-Supervised Graph Neural Networks for Improved Electroencephalographic Seizure Analysis, ICLR2022
> > > > > >
> > > > > > **Q2. The claim that this is the first paper to use truly dynamic graphs on EEG do not seem to match the notation used in equations 1 and 2**.
> > > > > > Thank you for your comment, we have updated Section. 3.1 (lines 168-170).
> > > > > >
> > > > > >
> > > > > > **Figure 1 doesn't seem to make this distinction clearly either**.
> > > > > > We have now updated Figure 1.
> > > > > >
> > > > > > Here, we really thank you for your discussion. Hope our clarification addresses your concern.

---

> > > > > > > ### Comment · Reviewer_C4vK · 2024-11-28
> > > > > > >
> > > > > > > I thank the authors once again for their time. With regards to the focus on the theoretical contribution and differences/similarities with Gao and Ribeiro's work, that is clearer to me now. Some key points over the past days that were important (at least for me) to better understand key statements of this paper don't seem to be in the paper, so I'd suggest the authors to include some clarification points in the final version of the paper (this is more of a personal suggestion, not exactly important for this rebuttal discussion).
> > > > > > >
> > > > > > > I appreciate the time the authors took, once again, to try to clarify the points regarding the "truly dynamic" graphs. Lines 168-170 are in-line with the discussions we had up until now, but what I mentioned in my last comment was specifically about equations 1 and 2 in the paper which do not seem to reflect the claims made by the authors overall in the paper and in this rebuttal period.

---

> > > > > > > > ### Author Response · Authors · 2024-12-01
> > > > > > > >
> > > > > > > > We greatly appreciate your time and suggestions.
> > > > > > > > We also thank you for recognizing the goal of this paper and our contributions.
> > > > > > > > Due to the policy regarding PDF updates, we will further finalize the discussed points in the final version.
> > > > > > > >
> > > > > > > > For Equations (1) and (2), we understand that you may suggest omitting the lower $t$ since the adjacency matrices do not vary across time in previous EEG studies.
> > > > > > > > We interpret $A$ as having $t$ but being fixed and sharing the same structure across all $t$, which prevents it from effectively reflecting dynamics.
> > > > > > > > We have reconsidered the adjacency matrices and incorporated
> > > > > > > > $t$ explicitly. For this reason,
> > > > > > > > $t$ must remain in Equations (1) and (2), and our Lemma and Theorem rely on adjacency matrices that include $t$.
> > > > > > > > This is also why, in our last response, we further updated lines 168–170 in Sec. 3.1: "existing work constructs $A$ as fixed across $T$...".
> > > > > > > >
> > > > > > > > Here we sincerely thank you again for helping us clarify and improve this paper.

---

> > > > > > > > > ### Comment · Reviewer_C4vK · 2024-12-02
> > > > > > > > >
> > > > > > > > > Dear authors, thanks for the further clarification, the inclusion of the variable "t" in equations is now clear to me. I'm inclined to increase my score as I believe you clarified most of my and the other reviewers concerns. I'll wait until the final deadline in case the reviewers still answer your rebuttals. Thanks for all the time spent tackling my questions and clarifying my doubts throughout the rebuttal period.

---

> > > > > > > > > > ### Author Response · Authors · 2024-12-03
> > > > > > > > > >
> > > > > > > > > > Dear reviewer, we really appreciate your recognition, time, and thoughtful comments. We are very happy that our efforts have addressed your concerns, as well as similar concerns raised by other reviewers. Thank you so much for your valuable feedback and support.

---

### Official Review · Reviewer_TUaH · 2024-11-13

**Soundness:** 3
**Presentation:** 3
**Contribution:** 2
**Rating:** 6
**Confidence:** 4

**Summary:**

The paper introduces EvoBrain, a dynamic graph neural network (GNN) model designed to improve seizure detection and prediction using multi-channel EEG data. EvoBrain adopts a time-then-graph approach, sequentially modeling the temporal dynamics of EEG signals before applying a GNN to capture evolving spatial relationships across EEG channels. Key contributions include a theoretical expressivity analysis that suggests potential advantages of the time-then-graph approach, modest empirical gains in AUROC and F1 scores over existing graph-then-time and time-and-graph models, and a dynamic graph structure designed to better reflect changing brain connectivity.

**Strengths:**

- The paper provides a practical demonstration of time-then-graph modeling in the context of EEG-based seizure prediction, showing how this approach can effectively capture temporal graph representations. This comparison with time-and-graph and graph-then-time methods highlights its strengths in modeling dynamic changes in EEG signals.
- Clinical analysis demonstrated the explainability of the time-then-graph approach by showing learned temporal dynamics through constructed dynamic graphs, providing valuable insights for exploring potential biomarkers in brain connectivity associated with seizures.

**Weaknesses:**

- This work applies the time-then-graph approach (Gao & Ribeiro, 2022) to multi-channel EEG data and demonstrates similar positive outcomes, suggesting that the study may take a somewhat naive approach by primarily focusing on adapting time-then-graph to EEG data.
- Much of the theoretical grounding regarding the expressiveness of time-then-graph is already covered in (Gao & Ribeiro, 2022). Additionally, the EvoBrain architecture borrows the GRU-GCN model from that work, raising questions about whether any theoretical advancements in this study are specifically tailored to EEG data.
- The study employs time-then-graph for improved temporal dynamic learning but processes EEG data in the frequency domain using FFT. This choice may limit the representation of temporal signals, as FFT primarily captures stationary frequency content rather than transient temporal dynamics. Although FFT helps in identifying frequency-domain characteristics relevant to seizures, such as specific frequency bands (e.g., delta, theta), it may miss essential non-stationary temporal nuances critical for comprehensive seizure prediction. Using wavelet transforms, which capture both time and frequency, could potentially offer a more robust temporal representation, especially when using the time-then-graph approach.
- While identifying potential biomarkers through clinical analysis is valuable, it remains unclear what insights the collaborating neurosurgeon contributed. For instance, did the findings align with known biomarkers in classical EEG-based seizure research? Or did the identified regions have a functional link to seizure impact? Providing a deeper analysis on these points would enhance the clinical relevance of the findings.
- The Related Works section could better support this study's contributions by restructuring its three paragraphs with clear subheadings and transitions. This would enhance cohesion, clarify the connection to prior research, and position the current work more effectively within the broader literature.

**Questions:**

- (Gao & Ribeiro, 2022) already established the expressiveness relationships among graph-then-time < time-and-graph < time-then-graph using similar approaches with 1-WL GNNs. Does this paper present any novel theoretical aspects compared to the prior work or specific to EEG data? The same question applies to the computational complexity analysis in Appendix C—does this study offer more than a rephrasing of the findings in (Gao & Ribeiro, 2022)?
- The GRU-GCN model proposed in (Gao & Ribeiro, 2022) is used as EvoBrain in this study; were there any methodological contributions made to adapt this model specifically for EEG in the time-then-graph approach, or was it used as-is?
- Given that time-then-graph aims to prioritize temporal dynamics, why was the frequency domain chosen over the time domain? If time-domain input were used instead, would time-then-graph still outperform graph-then-time and time-and-graph?
- Have any previous seizure detection studies using EEG identified biomarkers similar to those found in this work?
- Based on the parameter counts in Table 3, I wonder if the relatively lower performance of the baseline models might be due to overfitting.
- Would the superior performance of the time-then-graph approach, as shown in Table 2, still hold if the baseline models were adjusted to have comparable layer and parameter count settings?
- In Figure 3, a clearer explanation would be helpful to distinguish between the static and dynamic graph structures used in prior methods and how they were integrated into each approach. For instance, is the static graph $(\mathbf {X},\mathbf {A})$ constructed from frequency spectrum features at each snapshot, as described in Section 4.2? Does the dynamic graph refer to GRU-derived graphs $(h^{node}, h^{edge})$?
- In Figure 5, the constructed dynamic graph remains somewhat ambiguous. Does it refer to $h^{edge}$ learned at each time step via $\text{GRU}^{edge}$?

Minor Typo:
- In Definition 1, "approache" $\rightarrow$  "approach"

---

> ### Author Response · Authors · 2024-11-25
>
> We appreciate your time and the recognition of the strength of our work.
>
> We have marked the revision in the color **orange** on the paper.
>
> **W1. W2. and Q1.  The study may take a somewhat naive approach by primarily focusing on adapting time-then-graph to EEG data. Does this paper present any novel theoretical aspects compared to the prior work or specific to EEG data?**
> While we arrive at the same conclusion, the proof differs partially and is approached from different perspectives.
> We have updated and clarified this in the Introduction at lines 40-46 and 83-86.
> Epilepsy is fundamentally a network disease, where abnormal connections that switch from a stable (non-seizure) state to an unstable (seizure) state across multiple EEG channels may serve as a more effective marker [1] (line 40-43).
> Therefore, our goal is to capture how such channel connections (formulated as a graph) change over time (temporal dynamics).
> The theoretical expressivity analysis presented in Appendix A.2 of the original paper provides a general proof for unattributed or edge-only graphs, evaluating the model's ability to distinguish graph structures based solely on their connectivity, without considering node features.
> In contrast, in the case of EEG graphs, the structure construction is based on **node features** (line 83-86), specifically the similarity measures between features of two EEG electrodes.
> As outlined in line 43-46, the node embeddings and similarity measures are key factors in determining the graph's construction.
> In Lemma 2 and a proof in Appendix B.1, we evaluate the necessity of incorporating both node and edge information to effectively distinguish dynamic EEG graphs for modeling seizures.
> Overall, this paper is the first to provide a theoretical analysis of EEG-oriented graphs that incorporates both nodes and edges.
> Our experiments highlight the necessity of using dynamic graphs as well as our theoretical finding, i.e., time-then-graph.
>
> [1] Adam Li et,al. Neural fragility as an eeg marker of the seizure onset zone. Nature Neuroscience,pp.1–10,2021.
>
> **Q2. Were there any methodological contributions made to adapt this model specifically for EEG**.
> Architecturally, we used the same GRU-GCN.
> As described in line 78-81,
> however, we investigated an appropriate adaption for plausible EEG modeling, specifically focusing on a dynamic graph construction described in Section 4.1.
> The brain exhibits rapid changes over short durations, so we compute different graphs for each time snapshot to represent such changes.
> This approach differs from existing methods by introducing dynamics to both node features (each channel EEG representations) and edges (structural changes), enabling the GRU to model dynamic temporal dependencies and pass these dynamics into the final GCN.
> Figure 3 shows the effectiveness and necessity of this adaptation.
> This paper lays the groundwork for modeling temporal-dynamic EEG seizure research.
> As outlined in line line 92, more advanced methods can be easily integrated and applied.
>
> **W3. and Q3. FFT may limit the representation of temporal signals. Why was the frequency domain chosen over the time domain?**
> The time-then-graph approach captures long-term temporal dependencies, such as 12s or 60s, as shown in our experiments.
> Frequency represents each short-time snapshot, e.g., 1s.
> Seizures are neurological events characterized by transient and abnormal brain activities that manifest as distinct waveforms and high-frequency oscillations in EEGs.
> We employ FFT on short snapshots to filter out noise from raw EEG signals and incorporates both frequency and time to model changes in frequency over time.
> In general, frequency features is crucial for representing EEG patterns, such as alpha waves, sleep spindles, and other characteristic features.
>
> **W4. and Q4. It remains unclear what insights the collaborating neurosurgeon contributed. Have any previous seizure detection studies using EEG identified biomarkers similar to those found in this work?**.
> Clinically, abnormal waveforms or connections in EEGs originating from specific brain regions, i.e., seizure onset zone (SOZ), contribute significantly to seizure detection.
> However, there is no clinically validated biomarker for SOZ.
> We visualized the temporal evolutions of different EEG snapshots (e.g., 1 second to 3 seconds or to 10 seconds) to assess how brain connections differ during various states and how these connections consistently change over an EEG.
> We then collaborated with neurosurgeons to evaluate whether these visualizations provide meaningful insights, such as whether the observed differences imply clinical significance or are merely variations without diagnostic value.
> Our results align with neuroscientific findings and we have updated Section 5.3 (line 519-525) in the revision.

---

> ### Author Response · Authors · 2024-11-25
>
> **W5. The Related Works section could better support this study's contributions by restructuring its three paragraphs with clear subheadings and transitions**.
> Thank you for your comments. We have revised it.
>
> **Q5. The relatively lower performance of the baseline models might be due to overfitting**.
> For most baselines, we set the baseline model's parameters to the optimal values proposed in their original papers.
> In fact, the results are nearly identical to those reported in [2].
> For BIOT, since it is a large model, we conducted parameter sensitivity experiments on the number of layers and embedding dimensions.
> We have updated these experiments in Appendix E.
> Overall, we believe the results align well with our theoretical analysis.
>
> [2] Siyi Tang et al., Self-Supervised Graph Neural Networks for Improved Electroencephalographic Seizure Analysis, ICLR2022
>
> **Q6. the baseline models were adjusted to have comparable layer and parameter count settings?**.
> As described in Appendix D, the baseline methods either follow their original official implementations or are adjusted to match the same layer and parameter settings as our method.
>
> **Q7. Explanation between the static and dynamic graph structures in Figure 3.**.
> Thank you for your comments. We have updated the caption of Figure 3 to clarify the differences between these graph structures.
>
> **Q8. In Figure 5, the constructed dynamic graph remains somewhat ambiguous**.
> Yes, it visualizes elements with high value in $h^{edge}$.
> We have updated this point in line 511-513.

---

> > ### Comment · Reviewer_TUaH · 2024-12-03
> >
> > Thank you for providing detailed responses to my questions. I appreciate the effort in addressing most of the concerns I raised, and the explanations of the experimental results are now much clearer.
> >
> > As I understand it, this paper aims to apply the theoretical framework proposed by (Gao & Ribeiro, 2022) to EEG-based dynamic graphs. Rather than introducing a new theory, it demonstrates the efficacy of the existing framework through experiments tailored to this specific application.
> >
> > Additionally, regarding **Q5** and **Q6**, while I suggested adjusting baseline hyperparameters to have a similar number of model parameters as EvoBrain instead of focusing on hyperparameter similarity, the additional experiments in Appendix E seem to indirectly address this concern.
> >
> > Given these efforts and the improvements made to the paper, I increased my rating.
> >
> > However, I noticed a potential discrepancy regarding the claim that (Gao & Ribeiro, 2022) provides a general proof only for unattributed or edge-only graphs. Upon reviewing their proof, it appears that node features $X$ are also utilized. I may have misunderstood the distinction, but could you clarify this further? Specifically, are the "node features" referenced in (Gao & Ribeiro, 2022) conceptually different from those in this paper? Additionally, some content from (Gao & Ribeiro, 2022) appears to overlap with what is presented here, which adds to the confusion. Clearly differentiating these aspects between the two works could help in better highlighting the unique contributions of this paper.

---

> > > ### Author Response · Authors · 2024-12-03
> > >
> > > Thank you for your feedback, recognizing our efforts and improvements, and the increased score.
> > >
> > > We would clarify their proof does not focus on node features.
> > > As noted in their Appendix (p.18), "Each snapshot is a Circular Skip Link graph with 7 **unattributed nodes**" and "$X^{(top)} = X^{(btm)} = 0$,"
> > > indicating that their analysis partially focuses on graphs without node features (i.e., $X$ = 0).
> > > As shown our introduction, node features are critical for EEG analysis.
> > > Because the structure construction is based on similarity measure between EEG nodes/channels (Section 4.1).
> > > We have updated the analysis to provide proofs that consistently incorporate node features (Appendix B.2).
> > > While the assumptions (i.e., whether node features are present) and the proofs differ, both works arrive at the same conclusion.
> > >
> > > Due to pdf update deadline, we will further improve this point based on the discussion from you and reviewer C4vK.
> > > Thank you for you discussion and suggestion.

---

### Meta-Review · Area_Chair_LrN7 · 2024-12-21

**Metareview:**

The paper introduces EvoBrain, a dynamic graph neural network (GNN) that uses a "time-then-graph" strategy for seizure detection and prediction from multi-channel EEG data. While it achieves competitive performance and provides theoretical insights, the novelty and methodological contributions specific to EEG signal processing remain insufficiently substantiated.

The strengths of the work lie in its practical application of dynamic graph modeling, computational efficiency, and translational potential for seizure prediction. The authors addressed many weaknesses through revisions, including providing additional experimental details and clarifying theoretical distinctions. However, key concerns persist. The paper's novelty claim is undermined by reliance on prior work (e.g., Gao & Ribeiro, 2022), with minimal architectural or methodological innovation. The claim of truly dynamic graph structures remains unclear, as prior methods also dynamically update graph adjacency matrices. Additionally, the choice of FFT for temporal modeling may inadequately capture non-stationary EEG dynamics. The lack of direct comparison to advanced GNN architectures and traditional machine learning models further limits the scope of evaluation. These unresolved issues leave questions about the work's significance and originality in the field.

**Additional Comments On Reviewer Discussion:**

During the rebuttal period, the authors addressed several points raised by the reviewers. Regarding the novelty of the "time-then-graph" approach, reviewers questioned the originality of the method due to its reliance on Gao & Ribeiro (2022) and prior work with dynamic adjacency matrices. The authors clarified theoretical distinctions, emphasizing unique aspects of handling node and edge features in EEG-based graphs and dynamically updating the adjacency structure at each timestep. They revised Sections 3.1 and 4.1 and added theoretical proofs to better articulate the novelty.

The lack of experimental variance and concerns about reproducibility were also highlighted. To address this, the authors conducted additional experiments with multiple random initializations, providing variance measures and updating Table 2 with standard deviations for key metrics. They further detailed hyperparameter settings and included sensitivity analyses in Appendix A3. Additionally, reviewers noted the absence of comparisons with traditional machine learning models. The authors added experiments comparing EvoBrain to SVM and Random Forest, demonstrating their model's superior performance, and updated Table 2 accordingly.

Reviewers also suggested including results from more advanced GNN architectures, as the initial evaluation was limited to simpler models. In response, the authors included a comparison with GRAPH4MER, a more complex GNN baseline, in Table 2. They clarified the distinction between "dynamic graph structures" and "dynamic graph modeling," revising equations and adding explanations to emphasize the evolving nature of their graphs.

The use of FFT for preprocessing raised concerns about its ability to capture non-stationary temporal dynamics. The authors justified FFT as effective for capturing short-term EEG features while acknowledging its limitations and suggesting wavelet transforms for future work. Concerns about clinical relevance and the role of neurosurgeons were addressed by enhancing Section 5.3 with a discussion on how the findings align with neuroscience research on seizure onset zones. They also clarified the neurosurgeons’ contributions to validating the results.

Lastly, the related work section was criticized for lacking structure and depth. The authors reorganized this section with clear subheadings and improved transitions to provide a more coherent discussion.

While the authors addressed many concerns with substantial clarifications and additional experiments, some reviewers remained skeptical about the extent of novelty and the claims regarding dynamic graphs. This is weighed higher than others in the final decision.

---

### Decision · Program_Chairs · 2025-01-22

Reject